# SWE-FFICIENCY: Can Language Models Optimize Real-World Repositories on Real Workloads?

**Jeffrey Jian Ma** [1]   **Milad Hashemi** [2]   **Amir Yazdanbakhsh** [2]   **Kevin Swersky** [2]   **Ofir Press** [3]   **Enhui Li** [4]
**Vijay Janapa Reddi** [1]   **Parthasarathy Ranganathan** [5]

## Abstract

Optimizing the performance of large-scale software repositories demands expertise in code reasoning and software engineering (SWE) to reduce runtime while preserving program correctness. However, most benchmarks emphasize what to fix rather than how to fix code. We introduce SWE-FFICIENCY, a benchmark for evaluating repository-level performance optimization on real workloads. Our suite contains 498 tasks across nine widely used data-science, machine-learning, and HPC repositories (e.g., `numpy`, `pandas`, `scipy`): given a complete codebase and a slow workload, an agent must investigate code semantics, localize bottlenecks and relevant tests, and produce a patch that matches or exceeds expert speedup while passing the same unit tests. To enable this how-to-fix evaluation, our automated pipeline scrapes GitHub pull requests for performance-improving edits, combining keyword filtering, static analysis, coverage tooling, and execution validation to both confirm expert speedup baselines and identify relevant repository unit tests. Empirical evaluation of state-of-the-art agents reveals significant underperformance. On average, agents achieve less than $0.23\times$ the expert speedup: agents struggle in localizing optimization opportunities, reasoning about execution across functions, and maintaining correctness in proposed edits. We release the benchmark and accompanying data pipeline to facilitate research on automated performance engineering and long-horizon software reasoning.

## 1. Introduction

Language models (LMs) are becoming an increasingly substantial part of software engineering, from LM-powered auto-complete to autonomous software-engineering agents that plan, implement, and verify changes in large repositories. Recent agentic systems show that LMs can fix functional bugs and implement small features (Jimenez et al., 2024; Jain et al., 2024b; Yang et al., 2024; Wang et al., 2025). However, most benchmarks for these systems focus on what gets fixed or resolved, not the properties of code implementations—overlooking runtime performance, memory efficiency, style, and other software-engineering concerns. As we reach the limits of hardware, software optimizations become critical and have tremendous impact: Jain et al. (2024a) show that pure software changes can reduce high-utilization workload throughput by 10% on Google's datacenter compute, saving estimated millions of dollars. Recent benchmarks begin to probe code performance (e.g., KernelBench, (Ouyang et al., 2025); PIE, (Shypula et al., 2024); EffiBench, (Huang et al., 2024)), but they avoid real-world, end-to-end workloads on real repositories. We therefore ask: *to what extent can LM agents optimize the runtime of real-world repositories on real-world workloads?*

Recent work has begun to evaluate whether LMs can improve repo-level software runtime, most notably, GSO (Shetty et al., 2025) and SWE-Perf (Fan et al., 2025): both benchmarks tackle the problem of repo-level code optimization. GSO provides each task with an oracle script verifying functional equivalence and runs hidden performance tests at evaluation. SWE-Perf adapts existing repo unit-tests for both correctness and performance measurement, with task instructions pointing agents to optimize specific functions. However, software repositories commonly separate correctness and performance tests (ISO/IEC, 2011), and immediate correctness oracles are usually unavailable. Thus, these setups still insufficiently assess a core part of performance engineering: investigating an unfamiliar repository to recover code semantics and correctness from the codebase alone. Performance engineers *characterize* a workload (which can be slow for any myriad of reasons); *localize* where to intervene; and, just as importantly, *localize tests*—identifying

---

[1]Harvard University [2]Google DeepMind [3]Princeton University [4]Xi'an Jiaotong University [5]Google. Correspondence to: Jeffrey Ma <jeffreyma@g.harvard.edu>.

*Proceedings of the 43rd International Conference on Machine Learning*, Seoul, South Korea. PMLR 306, 2026. Copyright 2026 by the author(s).

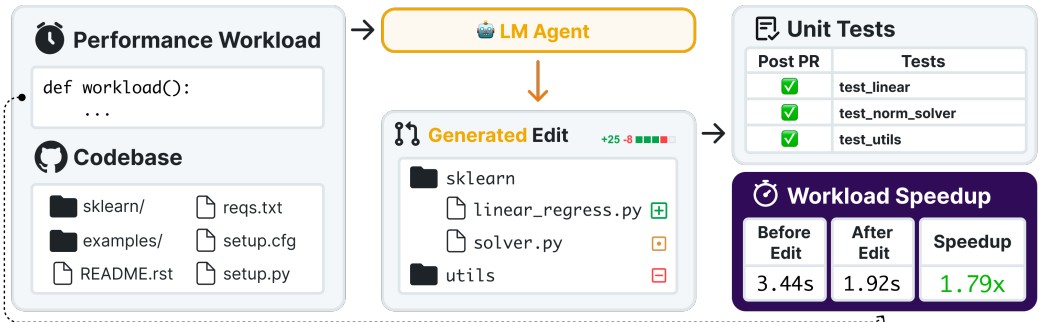

Figure 1. SWE-FFICIENCY evaluates the investigative, pass-to-pass workflow of performance engineering: given a codebase state and a targeted performance workload, agents must edit the codebase to speed up that workload while keeping relevant repo unit tests green.

and executing existing unit-tests to be confident that an optimization does not introduce new functionality. We design our benchmark to target this challenging and open-ended investigative workflow.

To address these gaps, we propose SWE-FFICIENCY (pronounced *swee-FISH-uhn-see*), a new benchmark to evaluate how well LMs can improve the performance of real-world workloads through modifying software repositories (Figure 1). To build SWE-FFICIENCY, we propose a novel, systematic data collection pipeline for optimization task instances, which uses attribute filtering, static analysis, code coverage, and execution validation. This anchors the realism of the benchmark and usefulness of the optimization tasks—a large and diverse set of 498 tasks across 9 codebases across data science, machine learning, and high performance computing. We score LM systems using *speedup ratio (SR)*, which evaluates how well models match or improve upon expert edits and motivates long-term progress on our benchmark. Analogous to how SWE-bench tests bug-fixing as a prerequisite skill for software engineering (rather than complete SWE workflows), SWE-FFICIENCY tests targeted workload runtime optimization as a foundational capability for performance engineering—a necessary skill before tackling more complex multi-objective regression management and production deployment. Furthermore, this setting itself is also realistic: expert engineers often accept specific workload optimizations in heavily used codebases (Appendix D).

We also conduct a holistic evaluation of LMs on SWE-FFICIENCY to better understand strengths and limitations. We reveal systemic gaps: on average, LMs achieve less than $0.23\times$ expert speedup and often introduce correctness bugs via proposed edits. Models struggle to localize the same expert optimization opportunities and prefer superficial speedups over more principled expert edits. While LMs exhibit promise in other SWE tasks, key advances in repo-level reasoning, systems optimization, and long-horizon planning are needed to close this expert gap.

**Our contributions.** (1) A diverse benchmark of 498 tasks across 9 repos, requiring deep codebase investigation and

test localization. (2) A scalable pipeline for extracting realistic and reproducible performance engineering tasks from GitHub repos. (3) An evaluation metric, speedup ratio, that measures parity with experts and encourages long term benchmark progress. (4) Empirical and qualitative analysis revealing large gaps between LMs and experts in edit localization and principled optimizations. (5) Open-sourced dataset, benchmark harness, and pipeline to accelerate research on automated performance engineering and long-horizon software reasoning.

**Conflict of Interest Disclosure.** Several authors of this work are employed by Google, which develops the Gemini family of language models evaluated in this paper (GEMINI 2.5 FLASH, GEMINI 2.5 PRO, GEMINI 3 FLASH, and GEMINI 3 PRO). To mitigate potential bias, all evaluations were run using identical agent harnesses, prompts, hardware configurations, and publicly available API endpoints across all models from all providers, and we report results for 15 frontier models spanning six organizations. The benchmark, evaluation harness, and model trajectories are released publicly to enable independent reproduction.

## 2. SWE-FFICIENCY Overview

SWE-FFICIENCY is a benchmark containing real performance-optimization GitHub pull requests from popular repositories. The task is to generate a pull request that modifies the codebase to make a given workload faster while preserving the correctness of existing repo tests.

### 2.1. Data Collection Procedure

We scrape nine popular Python GitHub repos, including `astropy`, `dask`, `matplotlib`, `numpy`, `pandas`, `scikit-learn`, `scipy`, `sympy`, and `xarray`. Building on SWE-bench infrastructure, we re-engineer the scraping pipeline to target performance-centric tasks (Figure 2). We designed filtering in Stages 2 and 5 to capture previously excluded performance edits, while adding important new stages for test coverage filtering and workload annota-

*Table 1.* SWE-FFICIENCY jointly (i) evaluates the runtime of performance workloads, (ii) verifies correctness using a repository's own tests, and (iii) uses separate correctness and performance workloads. For more details on related benchmarks, see Section 5.

| Benchmark | Evaluates Runtime | Repo Level | Correctness Eval: Using Repo's Own Tests | Performance Eval: *Separate* end-to-end system test | # of Optimization Tasks |
|---|---|---|---|---|---|
| SWE-BENCH | ✗ | ✓ | ✓ | ✗ | 0 |
| EFFIBENCH | ✓ | ✗ | ✗ | ✗ | 1000 |
| MERCURY | ✓ | ✗ | ✗ | ✗ | 1889 |
| PIE | ✓ | ✗ | ✗ | ✗ | 978 |
| KERNELBENCH | ✓ | ✗ | ✗ | ✗ | 250 |
| ALGOTUNE | ✓ | ✗ | ✗ | ✗ | 154 |
| GSO | ✓ | ✓ | ✗ | ✓ | 102 |
| SWE-PERF | ✓ | ✓ | ✓ | ✗ | 140 |
| SWE-FFICIENCY (OURS) | ✓ | ✓ | ✓ | ✓ | 498 |

tion (Stages 3 and 4). This ensures our pipeline identifies reproducible, verifiable optimization tasks rather than SWE-bench-style bug fix tasks.

**Stage I: Repo selection and instance scraping.** We target GitHub pull requests (PRs) from popular data science, machine learning, and high-performance computing repositories—these domains are performance-sensitive and contain PRs where authors explicitly optimize runtime. Widely-used libraries surface optimizations that are actually useful in real-world software.

**Stage II: Performance regression attribute filtering.** We prune away PRs that clearly are not performance-related or that introduce new behavior: in contrast, issue-resolution benchmarks like SWE-bench filter for the opposite by choosing tasks that add new tests. We select PRs only when (i) metadata includes performance keywords (i.e. `perf`, `speedup`, `benchmark`); (ii) PRs do not modify tests, to avoid behavior-changing edits misrepresenting as optimizations; and (iii) edits meaningfully modify the file's abstract syntax tree (AST), excluding no-op or docs-only diffs. This yields a distribution of code optimization tasks 100% disjoint from SWE-bench (see Appendix E.2).

**Stage III: Identifying covering correctness tests.** To enforce our benchmark's invariant specification (i.e. all relevant tests must continue to pass after an edit), we require

that at least one existing unit test exercises the modified code. We call these *covering* (or *intersecting*) tests: unit tests whose executed lines, or whose transitively-referenced symbols, intersect the lines, functions, classes, or modules modified by the expert patch. Note that this is a per-PR count, not the total size of the host repository's test suite. Per instance, we build a Docker image with pinned dependencies, run the repository's test suite under `coverage.py`, and combine the resulting dynamic line coverage with static `jedi`-based symbol resolution (using `ast`/`asttokens` to strip import- and definition-time coverage artifacts) to identify these covering tests. See Appendix E.3 for the full tool list and end-to-end pipeline.

**Stage IV: Annotating performance workloads.** Unit tests often do not capture runtime behavior, and software generally separates correctness from performance tests (ISO/IEC, 2011). Using PR descriptions and discussion as context, we manually annotate each task—writing a workload script that, when run before and after the PR's edit, shows a measurable performance improvement. Although PR info often includes ad-hoc demo scripts, these are not reliably auto-extractable; likewise, LM-based workload generation from patches, under the settings tested here, fails to consistently elicit claimed gains (see Sec. 4.2).

**Stage V: Execution-based filtering.** To curate a final set of

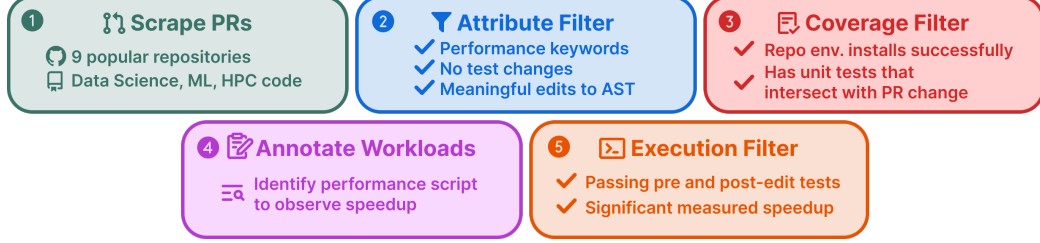

*Figure 2.* SWE-FFICIENCY collects tasks through a multi-stage scraping pipeline: each stage prunes candidate tasks that introduce new behavior, are unlikely to be performance related, or unsuitable for reproducible benchmarking. This yields a set of tasks, each of which have an accompanying expert or *gold* patch. See Appendix E for stage-specific details.

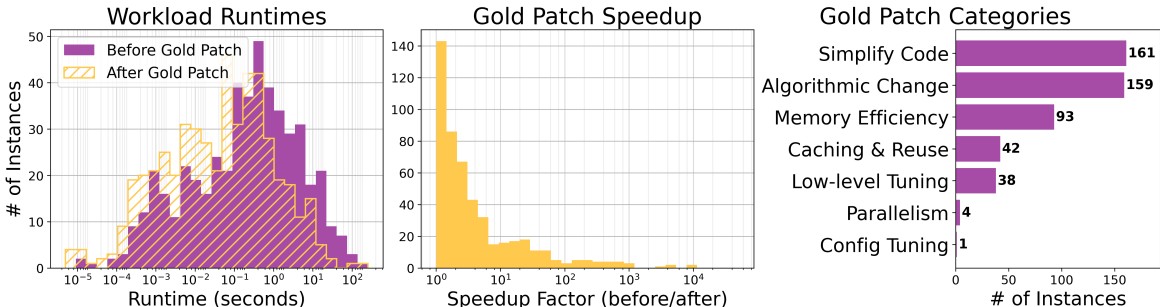

*Figure 3.* SWE-FFICIENCY contains a diverse distribution over performance workload runtime (left); over gold patch speedup (speedup achieved from expert PR edit); and over types of optimizations made by the expert (right). We use an LM to categorize the gold patch for each instance (for high-level analysis only) and manually verify a randomly chosen subset: see Appendix B.

verifiable and consistently reproducible optimization tasks, we run each instance's unit tests and annotated workload in a controlled environment (containerization, resource pinning) to ensure no interference with speedup measurements. We retain only instances that demonstrate significant speedups (runtime improvement greater than $2\times$ measurement std. dev.) and log test statuses for benchmark correctness checks. We separately verify post-curation that Speedup Ratio is stable to $< 0.5\%$ across independent evaluation runs (Appendix N).

### 2.2. SWE-fficiency Dataset Distribution and Unique Benchmark Features

**Open-ended but precise evaluation criteria.** Providing agents with just an executable code snippet (workload) and codebase makes the task very open-ended: agents can choose any approach, including changes different from the expert's *gold* patch. This mirrors the flexibility of real-world performance engineering—there rarely is a single prescriptive path to faster implementations. Our evaluation is made precise by grounding correctness in a set of unit tests and measuring LM speedup against gold patch speedup (i.e. how much speedup the expert achieved). Our benchmark encourages creativity in edit strategies while guaranteeing unambiguous criteria for strong optimizations.

**Clear distinction between performance and correctness tests.** Repositories generally require unit tests to run quickly (unlike more substantial performance workloads), and software standards encourage performance benchmarks to be clearly separated from correctness tests (ISO/IEC, 2011). Thus, defining a performance workload using unit-tests or a combined correctness-performance oracle is not fully reflective of actual performance engineering. Instead, SWE-FFICIENCY clearly separates performance evaluation workloads from repo correctness tests.

**Preserving existing correctness during optimization.** Unlike SWE-bench's issue-resolution setting (evaluating

bug-fixes that flip failing tests to passing), our benchmark targets *pass-to-pass* optimization—speeding up already-correct code without introducing new behavior. Specifically, we choose PRs that do not introduce new behavior: edits that introduce new features (and new tests) may have unintended performance effects on other workloads, and confound our specific evaluation of code optimization abilities. Evaluating how agents perform edits in this constrained task provides more confidence they can be deployed in real codebases without disrupting existing behavior.

### 2.3. Task Formulation

**Model input.** An agent is given a complete codebase and a performance workload exercising codebase functionality. We task the agent with modifying the codebase so that workload runtime improves while expected repository unit tests still pass. Expert performance engineers only require a reported slow workload and codebase to start optimizing: first characterizing the workload's bottlenecks, modifying files, verifying speedup against the workload and identifying relevant unit tests to check for no regressions. For examples of performance workloads, see Appendix B.2.

**Evaluation metrics.** Our evaluation metric is *speedup ratio (SR)*, which answers the question: *normalized to the expert edit, how well does the LM's generated edit perform?*. We apply an LM's submitted patch to the codebase and run repository test files associated with each instance. If the patch applies successfully and all tests pass, we compute the instance *speedup ratio* as $SR = Speedup_{\text{LM}}/Speedup_{\text{gold}}$ where gold speedup is $Speedup_{\text{gold}} = T_{\text{pre}}/T_{\text{post-gold-patch}}$ and LM speedup is $Speedup_{\text{LM}} = T_{\text{pre}}/T_{\text{post-LM-patch}}$. For example, if the expert achieves a gold speedup of $5\times$ and the LM achieves $1.2\times$ on the same instance, $SR = 1.2/5 = 0.24$. To aggregate across instances, we take the *harmonic mean* of each task SR, applying a per-instance lower bound of $SR_{\text{min}} = 0.001$ so that patches with near-zero speedup ratios do not dominate the harmonic mean and obscure differences in overall capability. If a system submits an empty

patch or a patch that fails unit tests, the instance's speedup ratio is similarly clamped to $\max(1/Speedup_{\text{gold}}, SR_{\min})$. We use harmonic mean since it is most appropriate for averaging speedup ratios (Smith, 1988; Eeckhout, 2024).

**Why a factor-based evaluation metric (not % solved)?** We adopt speedup ratio because it provides a long runway for progress and explicitly rewards surpassing experts. Percentage-style metrics collapse to two regimes—near $0\%$ today and nearer $100\%$ once tests are routinely passed—leaving little room to compare systems once the benchmark hits saturation. Speedup ratio motivates *continued progress* and keeps the leaderboard competitive after models surpass experts. This means our benchmark stays meaningful for the community both now (when systems only reach $0.23\times$ of expert performance) and later (when models might consistently score above $1\times$).

## 3. Evaluation Setup

**Machine and Evaluation Configuration.** We containerize each task environment into Docker images for reproducibility and easy integration with coding agent harnesses. All evaluations are run on a single Google Cloud `n2-standard-64` VM (64 vCPUs, 256GB Memory). To parallelize the benchmark for faster evaluation without workers interfering with another, we pin each worker to an exclusive set of *physical* CPU cores (4 vCPUs), the CPU's corresponding memory node, and assigning a memory limit (16GB) per worker. We observe $< 0.5\%$ variance in Speedup Ratio across independent measurement runs on our fixed hardware platform, validating that our measurements are reproducible and stable (see Appendix F and N). Furthermore, our harness rejects submitted solutions that exploit "compute once, re-use forever" across timing runs or solutions that try to introspect the call stack to detect if in a evaluation context (Appendix L.1 and L.2).

**Agent Scaffold.** We provide baseline performance on two open-source agent harnesses, OPENHANDS (Wang et al., 2025) and SWE-AGENT (Yang et al., 2024). Both scaffolds provide file-editing tools and a bash terminal interface for LM agents to easily edit code and execute commands. We configure agents with a 3-hour time limit per task, a 30-minute timeout per step, a maximum action count of 100, and provide the same number of vCPUs and memory as the evaluation setting. The agent is provided a task prompt and repo-specific commands for rebuilding (to support possible C/C++/Cython edits) and executing arbitrary test files. OpenHands is run with no dollar-cost limit, but in the SWE-AGENT setting we configure the underlying LM with a $1 token-spend max per task to observe performance under limited inference cost. We provide evaluations on both harnesses specifically for CLAUDE-3.7-SONNET, GPT-5

MINI, and GEMINI 2.5 FLASH models. (see Appendix G).

**Models.** We evaluate 15 frontier models from OpenAI, Anthropic, Google, Z.ai, Moonshot AI, and DeepSeek. For each instance, we sample a single trajectory and report the aggregated speedup ratio. We focus on $pass@1$ because it best matches both agent capabilities and realistic human workflows: (i) expert pull requests are effectively $pass@1$ (infeasible to review multiple PR submissions) and (ii) agentic LMs can still explore multiple edits, execute workloads repeatedly, and iterate over alternatives within a single trajectory. This also follows common practice in prior benchmarks like HumanEval and SWE-bench, where $pass@1$ is the primary metric (Chen et al., 2021).

## 4. Experiments and Results

On SWE-FFICIENCY, leading LM agents trail experts and often introduce correctness bugs. Our benchmark enables key quantitative and qualitative observations about LM agent behavior, namely how models solve easier cases, falter on harder ones, and exhibit *convenience bias* (a tendency to make small, input-specific, harder-to-maintain edits that exploit easy local wins instead of the principled structural changes experts pursue), underscoring the gap to expert-level performance engineering. All reported speedup ratios are stable to $< 0.5\%$ variance across independent runs on our fixed hardware (Table 11), so observed ranking differences between systems reflect genuine capability gaps and not measurement noise.

### 4.1. Overall Performance

**Leading agents struggle on SWE-FFICIENCY.** Across all agents, we observe that LM agents struggle to achieve more than $0.23\times$ of expert level performance. Table 3 summarizes speedup ratio performance of leading software-engineering agents on SWE-FFICIENCY. We see a substantial capability transfer gap: GPT-5 MINI (OPENHANDS) achieved $0.019\times$ of expert speedup, while the same system scored 62.6% on SWE-bench Verified. This indicates that current agents, while successful on issue-resolution and bug-fix tasks, currently do not immediately transfer to efficiency-oriented program changes, showing substantial headroom for improvement.

**Agents often introduce bugs during optimization.** LM agents often propose edits that cause repository unit tests to newly fail, invalidating any optimizations made. Table 3 unpacks agent performance across different unit test and performance outcomes: even when patches are functionally correct, the majority of edits are still slower than the expert. Strikingly, with the exceptions of GPT-5, CLAUDE 4.1 OPUS, and CLAUDE 4.5 SONNET, fewer than a quarter

*Table 2.* SWE-FFICIENCY results (higher is better). Human-expert SR is $1.0\times$. All experiments used OPENHANDS.

| System | Speedup Ratio (↑) |
|---|---|
| CLAUDE 4.5 OPUS | $0.225\times$ |
| GPT-5 | $0.157\times$ |
| GPT-5.2 | $0.148\times$ |
| CLAUDE 4.5 SONNET | $0.116\times$ |
| GEMINI 3 FLASH | $0.106\times$ |
| GEMINI 3 PRO | $0.102\times$ |
| CLAUDE 4.1 OPUS | $0.098\times$ |
| QWEN3 CODER PLUS | $0.068\times$ |
| KIMI K2-0905 | $0.054\times$ |
| DEEPSEEK V3.1 | $0.043\times$ |
| GLM-4.6 | $0.042\times$ |
| GEMINI 2.5 FLASH | $0.041\times$ |
| GPT-5 MINI | $0.039\times$ |
| GEMINI 2.5 PRO | $0.031\times$ |
| CLAUDE 3.7 SONNET | $0.024\times$ |

*Table 3.* Breakdown of patch outcomes. "Passes correctness" refers to functional correctness. Pre-edit denotes codebase before any edits.

| System | Fails Tests (↓) | Passes Correctness Tests | | |
|---|---|---|---|---|
| | | Slower than Pre-edit (↓) | Faster than Pre-edit (↑) | Faster than Expert (↑) |
| CLAUDE 4.5 OPUS | 9% | 1% | 47% | 43% |
| GPT-5 | 18% | 4% | 32% | 46% |
| GPT-5.2 | 11% | 3% | 34% | 52% |
| CLAUDE 4.5 SONNET | 19% | 5% | 44% | 33% |
| GEMINI 3 FLASH | 25% | 5% | 33% | 37% |
| GEMINI 3 PRO | 14% | 3% | 36% | 47% |
| CLAUDE 4.1 OPUS | 15% | 4% | 43% | 38% |
| QWEN3 CODER PLUS | 18% | 13% | 44% | 25% |
| KIMI K2-0905 | 26% | 11% | 44% | 19% |
| DEEPSEEK V3.1 | 19% | 18% | 44% | 18% |
| GLM-4.6 | 33% | 13% | 38% | 16% |
| GEMINI 2.5 FLASH | 39% | 12% | 38% | 11% |
| GPT-5 MINI | 45% | 12% | 30% | 13% |
| GEMINI 2.5 PRO | 40% | 18% | 34% | 8% |
| CLAUDE 3.7 SONNET | 35% | 11% | 33% | 21% |

of solutions are both correct and outperform expert-level speedups.

**Strong on easy wins, weak on harder speedups.** We identify three measures of task difficulty: (1) *pre-edit workload runtime* (longer duration workloads likely require more algorithmic insight); (2) *gold patch length* (harder instances require editing more lines); and (3) *the speedup factor* that the expert edit achieves (instance is harder if expert speedup is larger). Figure 4 shows a breakdown of benchmark performance in relation to these task difficulty measures. Across all three measures of task complexity, LMs are able to match expert performance on lower-complexity tasks. However, LMs struggle to solve tasks with longer duration workloads or larger feasible speedup opportunities. Interestingly, benchmark performance is largely unchanged as expert patch size increases, suggesting that core under-performance comes from reasoning about where to intervene, and not how large the intervention is.

**Function-level mislocalization severely limits LM performance.** Much of LM underperformance appears to stem from *failing to optimize the same functions as the expert*. If we view expert (gold) speedup as "mass" distributed over edited files and functions, Figure 6 shows that over *68% of expert gains occur in functions the LM never edits*. Although LM and expert modify the same files over 55% of the time, they miss the functions carrying most of the expert's speedup. Likewise, Figure 7 visualizes a function-level breakdown where the LM makes an attempted optimization in a deeper function and fails to match the expert's improvement. For more details, see Appendix I.

### 4.2. Qualitative Analysis

**LMs make satisficing optimizations, giving up before expert parity.** Figure 5 shows that the shortest sequences of agent actions (i.e. file-editing, running scripts) happen when LMs achieve speedup ratios exceeding $1\times$—expert-level wins are found early. When LMs underperform experts, median trajectory action counts sit at less than mid length (30–50 turns), well below the 100 action cap. This pattern fits a *satisficing* story (accepting the first measurable speedup and stopping, rather than searching for the deeper one an expert would find): once the model secures a measurable speedup, it tends to stop instead of pushing any closer to expert parity. Future agents can employ "don't-stop-early" triggers when code heuristics show larger possible speedups.

**Shortcut bias: caches and fast paths instead of structural cost reduction.** LMs preferentially add localized shortcuts—identity checks, ad-hoc early exits, and memoization—such as self-equality fast paths or persistent caches. Experts instead restructure code to reduce per-element cost. Figure 8 shows a flamegraph both after an LM versus an expert edit, where the expert optimizes by keeping work in fast `Arrow` kernels and producing a `BooleanArray` from a values/mask pair without materializing slow `object`-dtypes. Experts also use faster backends (Cython/Pythran/BLAS) to reduce Python overhead or remove Python-level work entirely—vectorizing, moving loops to compiled code, or dispatching to type-aware fast paths. LMs yield strong speedups only when these shortcut conditions hold, whereas systemic reductions are more broadly robust.

**Workload overfitting and semantic drift.** A second pattern is *workload overfitting*: LM patches hard-code properties of the specific evaluation workload (e.g., expected array shapes, dtypes, or index structures), achieving large speedups on that workload that would not survive minor input variations. At its worst, this crosses into outright correctness violations. For example, we observed agents return the input DataFrame unchanged from `groupby.apply`, or monkey-patch `np.arange` at the module level: changes that pass the timing workload but break unrelated callers. Experts instead target generalizable structure (i.e. multi-index skipping, per-dimensional slice reuse) while preserving functional behavior. During evaluation harness development, some agents exploited stackframe introspection to detect when code is being run in our evaluation environment or would try to cache computations across timing runs: we consequently added robust checks for this in the harness (see Appendix L.1 and L.2).

**Maintainability of generated edits.** LM edits are frequently invasive—global monkey-patching, module-level mutable caches, or fast paths tied to dynamic object attributes (an example shown in Fig. 9). Expert patches are localized and composable—adding a function call with precomputed constants, a Cython helper mirroring existing logic, or reusing shallow copies of constructor arguments. Expert edits have a lower blast radius of code edits and are more maintainable long term.

**Manually annotated workloads outperform LM generation.** Using our evaluation harness, we also study how well LMs can generate performance workloads. We compare the runtime improvement of each expert patch under two workloads: (i) an LM-generated (GEMINI 2.5 FLASH) workload produced from the gold patch and relevant files, and (ii) SWE-FFICIENCY's manually annotated workload (Stage 4, Fig. 2). Our annotations show stronger performance deltas 76% of the time, with 47% of LM workloads showing no significant speedups. Since performance engineering involves both bottleneck workload identification

and code optimization, we show how SWE-FFICIENCY can be further used to probe performance understanding in LMs. For more details, see Appendix K.

# 5. Related Work

**Function-level code efficiency benchmarks.** MER-CURY (Du et al., 2024), EFFIBENCH (Huang et al., 2024), and PIE (Shypula et al., 2024) quantify how often model-generated functions are slower than human references and study feedback- and goal-conditioned improvement. ECCO (Waghjale et al., 2024) emphasizes the necessity of correctness-preserving edits. Domain-focused work such as KERNELBENCH (GPU kernels; (Ouyang et al., 2025)) and ALGOTUNE (algorithmic redesign; (Press et al., 2025)) further stress wall-clock runtime as the metric. These function level efficiency benchmarks hold utilized libraries fixed and modify user-level workloads to be more efficient. We note that SWE-fficiency instead focuses on a "library maintainer" view of performance optimization, keeping the user workload fixed and editing the library to improve runtime.

**Repository-scale SWE benchmarks.** Benchmarks like SWE-BENCH (Jimenez et al., 2024; Yang et al., 2025), COMMIT0 (Zhao et al., 2024), and SWT-BENCH (Mündler et al., 2024) established that long-horizon reasoning over code repos is substantially harder than snippet tasks, but they mostly target fixing bugs, developing features and writing tests, rather than performance. Agentic systems (e.g. SWE-agent (Yang et al., 2024); OpenHands, (Wang et al., 2025)) supply the tooling to navigate, edit, run, and profile codebases, improving long-horizon outcomes.

**Repository-level performance datasets.** Closer to our setting, GSO (Shetty et al., 2025) and SWE-PERF (Fan et al., 2025) curate tasks from GitHub commits and evaluate repo-level runtime. SWE-PERF reuses repository unit tests for both correctness and performance and instructs agents to optimize specified functions; GSO employs both LM-generated correctness tests and performance workloads,

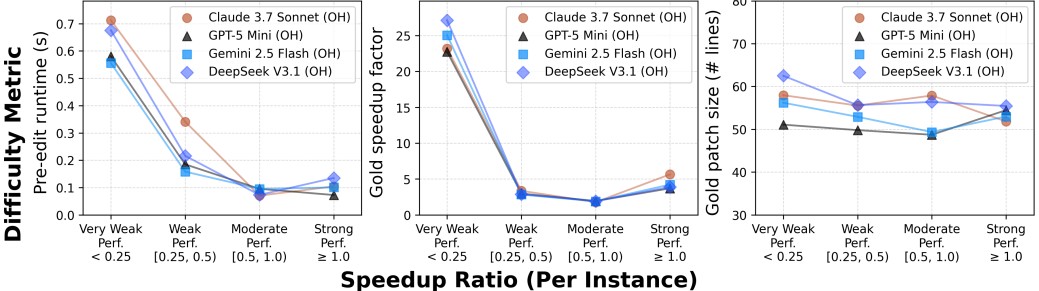

*Figure 4.* LMs achieve strong performance on easier problems but struggle on tasks with longer workload runtime duration and larger baseline expert speedups. We bucket LM submissions by per-instance speedup ratio and compute the geometric mean per-bucket of (i) pre-edit workload runtime, (ii) the gold (expert) patch speedup, and (iii) the number of lines in the gold patch.

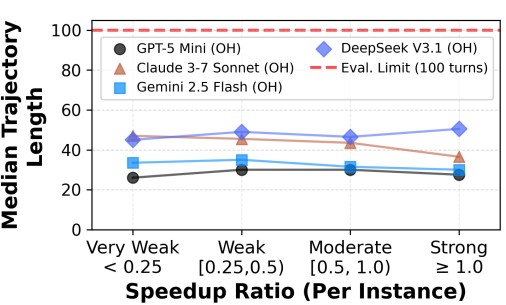

Figure 5. LMs find expert-level wins earlier on in action trajectories. When they underperform experts, LMs submit satisficing optimizations rather than trying on for expert parity.

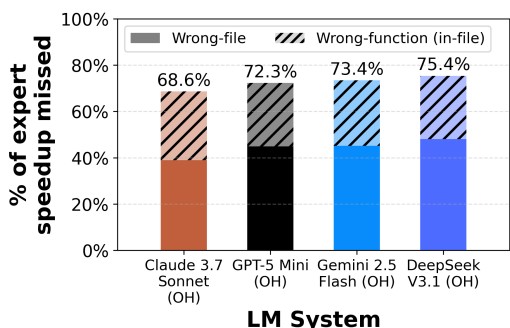

Figure 6. LMs leave a significant portion of expert-achievable speedup on the table due to wrong file/function selection and localization.

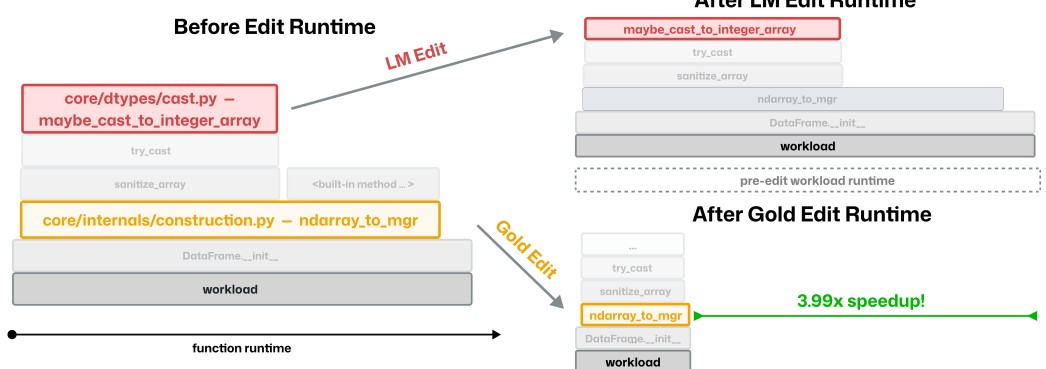

Figure 7. LMs prefer to edit different functions than the gold patch, missing out on major speedups. For a workload flamegraph for task `pandas-dev__pandas-52054`, CLAUDE 3.7 SONNET (SWE-AGENT) (red) chooses a different function (and file) than the expert (gold): it does not achieve the expert's overall workload speedup, since the expert's speedup is at a shallower scope.

providing a correctness oracle but not exposing performance workloads to agents. While suitable for measuring speedups, these designs insufficiently assess the *localization* skills central to performance engineering—*characterizing* a workload, *localizing* bottlenecks and edits, and *localizing tests* by discovering in-repo tests—capabilities we target directly.

## 6. Discussion

**Limitations.** SWE-FFICIENCY is primarily Python/-Cython changes across nine widely-used libraries; extending to lower-level stacks (C/C++/Rust) requires generalizing our coverage test selection and adding language specific build awareness. However, prior Python-only benchmarks (e.g. HumanEval, SWE-bench) have lead to accelerated progress in their respective research directions, and we believe that SWE-FFICIENCY can similarly motivate the research community. Finally, scaling to longer-running workloads, heterogeneous, non-CPU hardware, and multi-objective (multiple memory and runtime metrics) would further evaluate agent planning and measurement. Our curation and measurement methodology, prebuilt containers, performance isolation provide a solid foundation to build upon.

**Conclusion.** We present SWE-FFICIENCY, a repo-level benchmark of 498 optimization tasks across nine widely used repositories. Each task combines a performance workload, an expert patch with a significant speedup, and correctness tests covering the expert diff, enabling evaluation of *pass-to-pass* optimization. Our pipeline rigorously combines correctness, static analysis, coverage-guided test selection, manual workload annotation, and reproducibility checks. We present our metric, speedup ratio, for expert parity comparison and find that current agents remain well below expert performance. Our benchmark artifacts integrate with open agent frameworks, and we expose qualitative gaps with coding agents like mislocalization and shortcut bias. Our benchmark motivates long-term progress towards autonomous performance engineering and SWE agents.

## Acknowledgments

We extend our gratitude towards David Fleet and Deniz Altınbüken for reviewing the paper and providing insightful feedback. We also thank the extended team at Google DeepMind who enabled and supported this research direction. We gratefully acknowledge support from the Google Cloud

```
--- a/pandas/core/arrays/arrow/array.py          --- a/pandas/core/arrays/arrow/array.py
+++ b/pandas/core/arrays/arrow/array.py          +++ b/pandas/core/arrays/arrow/array.py
@@ -406,8 +406,14 @@ def _cmp_method(self, other, op):    @@ -406,8 +406,16 @@ class ArrowExtensionArray(OpsMixin,
-        result = result.to_numpy()                    ExtensionArray):
-        return BooleanArray._from_sequence(result)  -        result = result.to_numpy()
+        if result.null_count > 0:                    -        return BooleanArray._from_sequence(result)
+            values = pc.fill_null(result, False).to_numpy()  +        if result.null_count == 0:
+            mask = result.is_null().to_numpy()       +            result_np = result.to_numpy().astype(bool)
+        else:                                        +            return BooleanArray(result_np, np.zeros(len(result),
+            values = result.to_numpy()              dtype=bool))
+            mask = np.zeros(len(values), dtype=np.bool_)  +
+        return BooleanArray(values, mask)            +        result_np = result.to_numpy().astype(bool)
                                                      +        mask = result.is_null().to_numpy()
    def _evaluate_op_method(self, other, op, arrow_funcs):  +        return BooleanArray(result_np, mask)
        pc_func = arrow_funcs[op.__name__]
```

*Figure 8.* **Left:** Expert's edit (gold patch) on instance `pandas-dev__pandas-50524` optimizing a workload via avoiding a conversion to `object` dtype (20.5× speedup). **Right:** CLAUDE 3.7 SONNET (OPENHANDS) instead identifies a different fast path optimization when no null elements are present, but only achieves a 2.3× speedup (scoring a speedup ratio of 0.113×).

Research program, Gemini for Research program, and the Amazon Research Awards program for supporting evaluation runs on this paper. We also thank the Graham Neubig, Xingyao Wang, and the All Hands AI team for providing OpenHands runtime credits that enabled harness evaluations on this paper. Computations for this work were performed in part on the FASRC cluster supported by the FAS Research Computing Cluster at Harvard University. We thank Google, Anthropic, OpenAI, Alibaba Qwen, Moonshot AI, Z.ai, and the Harvard Data Science Initiative organizations for sponsoring credits and supporting model evaluations in our work.

## Impact Statement

This work introduces a benchmark for evaluating language model agents on real-world performance optimization of widely-used scientific Python libraries. We discuss broader impacts and data ethics in turn.

**Broader Impact.** Progress on automated performance engineering has the potential to reduce the compute and energy footprint of software that underlies much of modern data science and machine learning, and to lower the expertise barrier for performance optimization in open-source ecosystems. However, autonomous code-modification agents are dual-use: the same capabilities that produce speedups can also introduce subtle correctness regressions, maintainabil-

ity debt, or workload-specific shortcuts that fail to generalize. Our empirical analysis surfaces several such failure modes in current agents: *shortcut bias*, *workload overfitting and semantic drift*, and attempts at *reward hacking via stack-frame introspection and cross-run caching* (Section 4, Appendix L). SWE-FFICIENCY's pass-to-pass design, repo-grounded correctness tests, and harness-level reward-hacking mitigations are intended in part to make these failure modes visible during evaluation rather than after deployment. We encourage future work to retain these guardrails when deploying agent-generated optimizations into production library code, given the broad downstream user base of the repositories involved.

**Data Ethics and Release.** SWE-FFICIENCY is collected entirely from public repositories with permissive licenses (Table 4) and uses only data available via the public GitHub API; we do not collect information about pull request authors. Our work did not involve any human subject participation: we did not crowdsource or recruit human task workers for any part of SWE-FFICIENCY. For instance environment setup and annotation, the authors conducted all manual and semi-manual tasks.

We will open-source the task instances, evaluation infrastructure, and model trajectories, with documentation and community communication channels to support continued improvement of SWE-FFICIENCY.

```
--- a/pandas/core/series.py                       --- a/pandas/core/series.py
+++ b/pandas/core/series.py                       +++ b/pandas/core/series.py
@@ -1818,7 +1818,7 @@ def to_dict(self, into: type[dict] = dict)  @@ -1816,9 +1816,18 @@ class Series(base.IndexOpsMixin, NDFrame):
        -> dict:                                          else:
        else:                                     -            return into_c((k, v) for k, v in self.items())
-            return into_c((k, v) for k, v in self.items())  +            values = getattr(self, "_values", None)
+            return into_c(self.items())           +            if values is None:
                                                   +                return into_c((k, v) for k, v in self.items())
                                                   +            try:
                                                   +                list_vals = values.tolist()
                                                   +            except Exception:
                                                   +                # fallback to generic iteration
                                                   +                list_vals = [v for v in values]
                                                   +            return into_c(zip(self.index, list_vals))
```

*Figure 9.* **Left:** Expert's edit on `pandas-dev__pandas-50089`, optimizing `Series.to_dict` by replacing a key-value pair generator with a `items()` view, eliminating per-element tuple allocation (1.38× speedup). **Right:** GPT-5 MINI (OPENHANDS) converts the underlying array to a Python list, zipping with the index to reduce Python-level boxing when iterating (1.98× speedup).

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

## A. LLM Usage

Language models were used to polish writing, help with grammatical errors and typos, and to help check with compliance against ICML's author guide. Beyond the LM usage in our benchmark evaluations and experiments, they were not used in any other part of writing this work.

## B. Additional Dataset Information

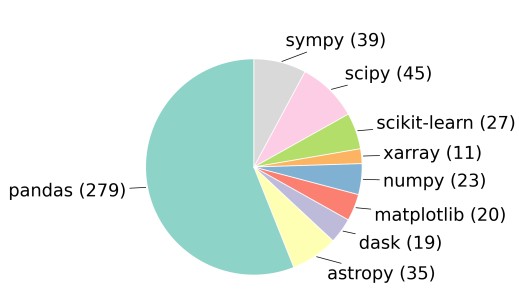

| Category | Metric | Mean | Max |
|---|---|---|---|
| Codebase | # Instances | 498 | |
| | # Repos | 9 | |
| Workload | # of Lines | 25.47 | 180 |
| | Runtime (s) | 4.47 | 257.09 |
| Gold Patch | # Lines edited | 49.1 | 2445 |
| | # Files edited | 2.2 | 12 |
| | Speedup | 2.64× | 249k× |
| Tests | # Covering Tests | 54k | 222k |

*Figure 11.* Additional summary statistics for SWE-FFICIENCY dataset. Arithmetic mean is used in all cases, except for speedup (harmonic mean). Test counts report *covering* (intersecting) tests per instance—unit tests whose dynamic line coverage or symbol-expansion intervals intersect the lines, functions, classes, or modules modified by the expert patch—not the total test count of the host repository.

*Figure 10.* Repository distribution of task instances in the SWE-FFICIENCY dataset.

In this section, we provide more details on the dataset summary and distribution of SWE-FFICIENCY. We verify that all repositories used have permissive licenses, allowing for data mining and inclusion into the SWE-FFICIENCY benchmark, as shown in Table 4. Figure 10 and 11 show additional details on the distribution of task instances, repositories, and corresponding information. We observe that compared to SWE-bench, SWE-FFICIENCY gold patches are larger on average and have significantly larger numbers of related tests (as checking for correctness regression is stricter than SWE-bench's pass criteria of issue resolution).

We also note that SWE-bench does not contain any code optimization tasks as noted in Table 1. Firstly, as we select for changes that do not introduce test file changes, this disqualifies any of SWE-bench instances from passing Stage 2 of Figure 2 (attribute filtering). Furthermore, we also run SWE-bench instances through our performance keyword pipeline and randomly sample 50 of the 581 resulting instances for human review: all of these instances had issue statements or hints text that clearly show the change introduces new behavior (which is tested by the SWE-bench instance's `test_patch`).

### B.1. Repository Descriptions and Permissive Licenses

All SWE-FFICIENCY task instances are scraped from public GitHub repos with permissive licenses via the public GitHub API as stated in our Ethics Statement. Repository specific licenses are shown in Table 4, where links to custom licenses are provided inline.

### B.2. Example Performance Workloads

We provide some examples of the performance workloads associated with each task instance. To recap, each performance workload consists of (i) necessary imports, (ii) an optional `setup` function, which sets up work that should not be runtime benchmarked, (iii) a `workload` function, which runs some repository functionality and runtime to be measured, and (iv) performance measurement harness code. Each problem runs the workload and setup multiple times to generate a distribution of runtimes, from which we compute the mean and standard deviation. During our dataset curation pipeline, we reject task instances and workloads that fail to show statistically significant improvements in the execution validation stage (Appendix E.5).

*Table 4.* SWE-FFICIENCY GitHub repositories, package description, and their permissive licenses.

| Repository | Description | License |
|---|---|---|
| astropy/astropy | Astronomy and astrophysics core library | BSD-3-Clause |
| dask/dask | Parallel computing library for analytics | BSD-3-Clause |
| matplotlib/matplotlib | Plotting and graphics library | Custom |
| numpy/numpy | Core scientific computing library | Custom |
| pandas-dev/pandas | Core data analysis library | BSD-3-Clause |
| pydata/xarray | Multi-dimensional array library | Apache 2.0 |
| scikit-learn/scikit-learn | Machine learning in Python | BSD-3-Clause |
| scipy/scipy | Package for math, science, and engineering | BSD-3-Clause |
| sympy/sympy | Computer algebra system written in Python | Custom |

**Performance Workload for `dask__dask-6293`**

```python
import timeit
import statistics

import dask
import dask.array as da
import numpy as np

def setup():
    global stacked, sub_arrays
    sub_arrays = [
        da.from_delayed(
            dask.delayed(np.zeros)((100000,), dtype="int64"),
            shape=(100000,),
            dtype="int64",
            name=idx
        )
        for idx in range(10000)
    ]
    stacked = da.stack(sub_arrays)

def workload():
    global stacked, sub_arrays
    for i in range(len(sub_arrays)):
        stacked[i]

runtimes = timeit.repeat(workload, number=1, repeat=5, setup=setup)

# Print runtime mean and std deviation.
print("Mean:", statistics.mean(runtimes))
print("Std Dev:", statistics.stdev(runtimes))
```

**Performance Workload for `numpy__numpy-11720`**

```python
import timeit
import statistics

import numpy as np

b = np.random.random((5, 2))
t = np.random.random((5, 5, 2))
p = np.random.random((2, 5))

def workload():
    out = np.einsum('ij,ixy,ji->xy', b, t, p)

runtimes = timeit.repeat(workload, number=100, repeat=10**4)
```

```
# Print runtime mean and std deviation.
print("Mean:", statistics.mean(runtimes))
print("Std Dev:", statistics.stdev(runtimes))
```

## Performance Workload for `scipy__scipy-12001`

```
import timeit
import statistics
import numpy as np
from scipy.stats import maxwell

def setup():
    global data
    data = maxwell.rvs(loc=2.0, scale=3.0, size=100000, random_state=42)

def workload():
    global data
    _ = maxwell.fit(data)

runtimes = timeit.repeat(workload, number=1, repeat=200, setup=setup)

print("Mean:", statistics.mean(runtimes[-100:]))
print("Std Dev:", statistics.stdev(runtimes[-100:]))
```

### B.3. LM Classification of Gold Patch Edit Types

To examine the diversity of gold (expert) patch optimizations that SWE-fficiency submissions are graded against (rightmost plot in Figure 3), we prompt GEMINI 2.5 FLASH with the following task prompt to extract optimization categories. We then randomly sampled 50 LM classifications for manual review and confirmed the high level categorization and explanation for each. We emphasize that during SWE-FFIENCY evaluation, LM agents are free to make any optimization desired to improve the performance workload, and we use this categorization strictly to show the diversity of our collected dataset.

## Performance Diff Classification Task Prompt

```
You are an excellent performance engineer. Given a code diff and an affected
↪  performance workload that shows a speedup as a result of this diff, output a
↪  single high-level performance bucket, the concrete signals from the diff that
↪  justify that bucket, a mechanism-level explanation of **why the specific code
↪  edits** improve performance, and a confidence score. Prefer software-side
↪  mechanisms; ignore hardware/microarchitecture unless explicitly cited.

## Inputs

* Performance workload and code diffs.

## Buckets (choose **exactly one** `classification`)

1. **Algorithmic / Data Structure Improvements** -- Better asymptotic complexity or
↪  more suitable data structures; removes redundant passes.
2. **Memory Efficiency & Management** -- Fewer allocations/copies; pooling/reuse;
↪  layout/locality changes; alignment; `reserve()`; SoA/AoS.
3. **Concurrency & Parallelism** -- Threading/async; work partitioning; lock
↪  scope/structure; atomics; SIMD/vectorization.
4. **I/O and Storage Efficiency** -- Fewer syscalls; buffering/batching; async I/O;
↪  (de)serialization changes; payload trimming.
5. **Code Simplification / Dead-Code Elimination** -- Early exits; pruning
↪  logging/asserts on hot paths; removing unnecessary work/branches.
6. **Compiler / Build / Low-level Tuning** -- Flags (LTO/PGO), inlining hints,
↪  intrinsics, UB fixes enabling optimization, branch hints.
7. **Configuration / Parameter Tuning** -- Constants/thresholds/buffer sizes,
↪  thread-pool/GC settings, feature flags altering performance behavior.
```

8. **Caching & Reuse** -- Memoization, caches, reuse of precomputed artifacts/results,
↪ avoiding repeated expensive calls.
9. **Unknown / Not Enough Information** -- Claimed speedup but mechanism not
↪ inferable from available changes.

## Secondary Tags (optional)

* Other relevant buckets or keywords, if any (e.g., "Memory Efficiency & Management"
↪ and "Caching & Reuse" could both apply).
* These can be more specific, e.g., "memoization", "lock-free", "buffered I/O", but
↪ try to use standard terms where possible.

### Disambiguation rules

* **Algorithmic vs Caching**: If an algorithm was fundamentally changed and a cache
↪ was added as a helper, choose **Algorithmic**; add `memoization` in
↪ `mechanism_signals`.
* **Concurrency vs Algorithmic**: If parallelism is added without changing the
↪ algorithm, choose **Concurrency & Parallelism**.
* **I/O vs Memory**: If copies were removed primarily to cut syscalls or shrink
↪ payloads, choose **I/O**; if focused on allocation/locality/pressure, choose
↪ **Memory**.
* **Compiler/Build**: Choose **Compiler / Build** when source logic is the same but
↪ flags/hints/toolchain changed.
* **Benchmark-only**: Choose **Workload/Benchmark-Only** when only
↪ harness/warmup/affinity/timers changed.

## What to extract as "signals"

Short, concrete phrases tied to the diff, e.g.:

* containers/algos: "`vector`→`unordered_map`", "removed nested loop", "added binary
↪ search", "streaming parse"
* memory: "added `reserve()`", "object pool", "moved to stack allocation", "SoA
↪ layout"
* concurrency: "introduced thread pool", "reduced lock scope", "lock-free queue",
↪ "SIMD intrinsics"
* I/O: "batched writes", "buffered reader", "protobuf→flat serialization",
↪ "compression level tuned", "fewer syscalls"
* simplification: "early return before parse", "pruned logging on hot path", "deleted
↪ dead branch"
* compiler/build: "enabled LTO/PGO", "added `inline`/`cold`/`likely`", "UB fix
↪ unblocking vectorization"
* config: "increased read buffer to 1MB", "thread pool size = cores"
* caching: "added LRU cache", "memoized function result"
* meta: "only bench harness changed"

## Output Requirements (STRICT)

* Output **only** the JSON object -- no prose, no Markdown, no code fences.
* Keep `explanation` <= 6 sentences and tie it to specific lines/files/patterns from
↪ the diff.
* If evidence is weak or ambiguous, use `classification: "Unknown / Not Enough
↪ Information"` and lower `confidence`.

### JSON Schema

```json
{
  "classification": "<one of the 10 buckets>",
  "secondary_tags": ["<optional: other relevant buckets or keywords, if any>"],
  "mechanism_signals": ["<short phrases pulled from the diff that justify the
  ↪ classification>"],
  "affected_components": ["<files/modules/functions inferred from paths/symbols>"],
```

```
    "explanation": "<mechanism-level rationale grounded in the diff: what changed, how
    ↪  it reduces work/contention/latency/allocs/syscalls, and why that maps to the
    ↪  chosen bucket>",
    "confidence": "<high|medium|low>"
}
```

## Final sanity check (do this before emitting JSON)

1. Have I picked **exactly one** bucket that best explains the performance mechanism?
2. Do my `mechanism_signals` cite concrete code changes from the diff that motivate
↪  that bucket?
3. Is the explanation mechanism-centric and grounded in the edits (not benchmarks)?

---

### Mini-examples

**A. Algorithmic / Data Structure Improvements**

```json
{
  "classification": "Algorithmic / Data Structure Improvements",
  "secondary_tags": ["asymptotic complexity"],
  "mechanism_signals": ["removed nested O(n^2) scan", "introduced unordered_set",
  ↪  "added reserve() to avoid rehash"],
  "affected_components": ["src/import/dedupe.cpp", "Importer::dedupeRecords"],
  "explanation": "The patch replaces a quadratic duplicate search with hash-based
  ↪  membership checks and preallocates the table to avoid rehash, removing repeated
  ↪  comparisons across the hot loop.",
  "confidence": "high"
}
```

**B. Concurrency & Parallelism**

```json
{
  "classification": "Concurrency & Parallelism",
  "secondary_tags": ["parallelism", "lock contention"],
  "mechanism_signals": ["introduced thread pool", "tile-based work partitioning",
  ↪  "narrowed mutex scope"],
  "affected_components": ["decoder/pipeline.cc", "decoder/tiler.cc"],
  "explanation": "Work is partitioned by tile across a shared pool and critical
  ↪  sections are reduced to short regions, lowering contention and enabling
  ↪  parallel execution of the same algorithm.",
  "confidence": "high"
}
```

**C. Unknown**

```json
{
  "classification": "Unknown / Not Enough Information",
  "secondary_tags": [],
  "mechanism_signals": ["broad refactor", "no evident hot-path edits"],
  "affected_components": ["loader/*"],
  "explanation": "Large refactor touches many files without showing hot-path changes
  ↪  or recognizable performance mechanisms, so the cause of any improvement cannot
  ↪  be determined from the diff alone.",
  "confidence": "low"
}
```

## B.4. Dataset Schema

For clarity, we describe the schema and description of our dataset in Table 5. We upload our dataset to HuggingFace (`swefficiency/swefficiency`) for easy community download, and refer readers to our code supplementary material to see how the dataset is directly used during evaluation.

*Table 5.* SWE-FFICIENCY dataset columns and description.

| Column Name | Description |
| --- | --- |
| `repo` | (str) Repository identifier for the task (e.g., `owner/repo` on GitHub). |
| `instance_id` | (str) Unique ID for this dataset instance. |
| `base_commit` | (str) Git commit SHA to check out before applying any patches; defines the baseline state under evaluation. |
| `patch` | (str) Expert git patch with source-code changes which solves the task and shows a performance optimization. For evaluation purposes, this should never be provided to the agent. |
| `created_at` | (str) ISO-8601 timestamp indicating when this instance was created. |
| `version` | (str) Repository version string for this instance (used for repository environment building). |
| `environment_setup_commit` | Commit SHA (or ref) that pins environment setup artifacts (e.g., dependency files) for reproducible evaluation. |
| `workload` | (str) Python workload script used for performance measurement (script/benchmark/entrypoint) See Appendix B.2 for examples.. |
| `test_cmd` | Shell command prefix used to run the test suite for this repo (e.g., `pytest -q`). |
| `rebuild_cmd` | Shell command to (re)build/reinstall the project between runs for agent usage (e.g., `pip install -e .`) |
| `image_name` | Container image tag/name providing the canonical evaluation environment (OS, toolchain, deps). |
| `covering_tests` | (list of str) List of tests paths that exercise the changed code regions (e.g., from coverage or curated mappings). For evaluation purposes, this should never be provided to the agent. |
| `single_thread_tests` | List of tests that must run serially (to avoid flakiness or resource contention) during evaluation. For evaluation purposes, this should never be provided to the agent. |
| `PASS_TO_PASS` | (list of str) List of test identifiers that pass after the expert edit and are expected to pass after an LM generated edit (regression guard). Specifically, these are the per-instance *covering* tests identified in Stage III—unit tests whose executed lines or transitively-referenced symbols intersect the expert patch—not the full repository test suite. For evaluation purposes, this should not be provided to the agent. |

## C. Why `timeit`?

Our benchmark uses Python's `timeit` module for performance measurement. We chose `timeit` because it reflects how experts actually measure performance in practice: examining the PRs in our dataset, we observe that PR authors themselves use `timeit` (or the `%timeit` IPython magic) to demonstrate speedups in their PR descriptions and discussions. Furthermore, `asv` (Airspeed Velocity)—the industry-standard benchmarking framework used by NumPy, SciPy, Pandas, and other major scientific Python projects—uses `timeit` under the hood for its timing measurements. This ecosystem validity is more important than theoretical purity: our benchmark measures performance the same way the community measures it.

We use the `timeit` Python API (specifically `timeit.repeat()`) rather than the command-line interface. This gives us access to the full distribution of runtimes across multiple repetitions, enabling statistical analysis of measurement variance. Each workload reports both mean and standard deviation, and we filter tasks during curation to ensure statistically significant speedups (see Section 2.1).

**Warmup and Steady-State Measurement.** A common concern in microbenchmarking is whether to discard initial "warmup" iterations to reach steady-state performance. We deliberately *do not* impose a universal warmup policy, instead preserving whatever warmup behavior (if any) the expert included in their original timing script. Furthermore, warmup is often employed due to difficulty in proper process isolation: we already account for this with our designed hardware isolation in Appendix F. This design choice reflects our core philosophy of mirroring real-world expert practice and measurement.

- **Fidelity to acceptance criteria.** Each task's ground truth is defined by the speedup that maintainers accepted when merging the PR. If the expert's demonstration did not include warmup iterations, imposing them post-hoc would change the acceptance criterion and potentially invalidate the task.

- **Preservation where appropriate.** Where experts *did* include warmup—particularly for JIT-compiled code (Numba, PyPy) or lazy-initialization patterns—our extracted workloads preserve this structure. The warmup is part of the workload's setup, not something we strip away.

We empirically validate measurement stability in Appendix N, showing $< 0.5\%$ coefficient of variation across independent runs for our curated tasks. Tasks exhibiting high measurement variance were filtered during curation (Section 2.1).

## D. Real-World Workload-Specific Optimization

A potential concern is whether optimizations that benefit only specific workloads (rather than all possible inputs) are realistic. We find strong evidence that experts routinely accept such workload-specific optimizations in production codebases, hence the validity of designing our benchmark towards this targetted runtime optimization skill.

- **numpy#21394**: This PR optimizes array operations specifically for small arrays. The optimization adds overhead for large arrays but was merged because small-array performance is a common bottleneck in real workloads.

- **pandas#44566**: This PR added computational overhead for short DataFrames in order to fix a performance regression on long DataFrames. The maintainers accepted the tradeoff because long-DataFrame performance was the more pressing production concern.

- **numpy#18324**: This PR showed no improvement on larger workloads but was merged because it addressed a specific bottleneck pattern that users encountered in practice.

These examples demonstrate that workload-specific optimization is standard practice in performance engineering. Experts optimize for the workloads that matter, accepting that some inputs may not benefit (or may even regress slightly). SWE-FFICIENCY's single-workload evaluation reflects this reality: the task is to optimize a *specific* slow workload, mirroring how performance engineers respond to user-reported bottlenecks.

## E. Additional Details on Data Collection Procedure

In this section, we provide more concrete details on the dataset collection procedure explained in Section 2.1 and in Figure 2. To recap, our dataset pipeline is comprised of the following stages (with numbers on the yield after each stage and direct references to appendix subsections):

1. Scrape pull requests from nine widely used open-source Python repositories in data science, machine learning, and high-performance computing—chosen for mature test suites and stringent performance requirements—yielding **96457 PRs** (Appendix E.1).

2. Filter for candidate performance-regression instances by requiring performance-related keywords, excluding PRs that modify tests (and introduce new behavior), and retain only edits that meaningfully change the abstract syntax tree (AST) — criteria that notably targets instances intentionally excluded by SWE-bench due to its test-change filter. After attribute filtering and prior to checking for meaning full changes to the AST, we retain **9257 PRs** (-90.4%) at this stage (Appendix E.2).

3. Construct an executable Docker environment per instance with manually curated, version-pinned dependencies (tests are often version-sensitive), run repository unit tests, and use line coverage to keep only instances with at least one "guarding" test whose executed lines intersect the edited code. We retain **1041 PRs** (-88.8%) up until this stage (Appendix E.3).

4. Annotate each surviving instance with a minimal workload script that reliably exposes the pre/post performance delta (Appendix E.4).

5. Run correctness tests and the performance workload before/after applying the patch, retaining only instances with a statistically significant improvement, recording post-patch test outcomes, and ensuring reproducibility via Docker with 4 CPU cores and 16GB RAM. This yields our final **498 tasks** across 9 repos (Appendix E.5).

## E.1. Repo selection and instance scraping

We provide more details on Stage 1 from Figure 2 in how we selected repositories and scraping raw instance information. In this stage, using **March 12th, 2025** as a cutoff date, we scrape merged pull requests from nine (9) repositories (`astropy`, `dask`, `matplotlib`, `numpy`, `pandas`, `scikit-learn`, `scipy`, `sympy`, and `xarray`). Note that we also relax the SWE-bench requirement that an valid PR require a linked GitHub issue as a problem statement: from our observation, we see that *many performance optimization PRs are opportunistic and do not always have a linked issue created ahead of time.*

We note that some repositories overlap with SWE-bench and SWE-Gym (Pan et al., 2025), such as `astropy`, `matplotlib`, `xarray`, `scikit-learn`, `sympy`, `dask`, and `pandas`, while others are exclusive to SWE-FFICIENCY like `numpy` and `scipy`. We examined all the repositories in SWE-bench and SWE-Gym to start and found that most of the repositories in those datasets that are not included in our benchmark contain very few performance related changes: This is due to some of those repositories focusing on general functionality rather than performance optimization specific changes. For example, packages like `flask` and `django` in SWE-bench focus on rapid iteration and high-level design rather than optimizing for high degrees of performance, which reflects in their list of merged pull requests (PRs) having very few occurrences of the word `perf`.

## E.2. Performance regression attribute filtering

As discussed in Section 2.1, we modify the attribute filtering from SWE-bench to prune away instances that are not regression-free performance optimization. Specifically, we keep an instance only if it satisfies the following:

1. *Does not contribute test changes:* We intentionally drop instances if they add test changes, since this indicates, with high likelihood, that new behavior is introduced by that PR. In contrast, SWE-bench selects for the opposite (intentionally selects for new tests, and masks out those tests for instance evaluation), meaning that *our dataset is exactly instance-wise disjoint* with both SWE-bench and SWE-gym. We attribute this to "test-driven development" in general: performance PRs generally do not change inputs or output behavior and thus do not need new tests to guard this new behavior.

2. *Contains performance related keywords or tags:* We check if the pull request metadata includes any of the following keywords: `performance`, `speedup`, `speeds up`, `speed-up`, `speed up`, `faster`, `memory`, `optimize`, `optimization`, `profiling`, `accelerate`, `fast`, `runtime`, `efficiency`, `benchmark`, `latency`, `throughput`, `multithreading`, `parallel`, `concurrency`, `concurrent`, `profiling`, CPU usage, `memory usage`, `resource usage`, `cache`, `caching`, `timeit`, and `asv`. We also check if pull request has been tagged with any repo specific performance tags and keep those pull requests as well.

3. *PR contains meaningful changes to AST:* We finally check that the PR edit has made meaningful changes to each changed file's abstract syntax tree (AST) as parsed by `tree-sitter`. This helps us ignore no-op changes that are comment or doc-string only and select more specifically for substantial performance related changes, which are almost guaranteed to require a modification to a code file's AST.

**Why SWE-bench-style pipelines miss performance PRs.** The core assumption underlying SWE-bench-style pipelines is that meaningful code changes should be accompanied by new tests. This holds for bug fixes and feature additions but fails for performance optimizations, which by definition do not change behavior—they make existing functionality faster without

altering inputs or outputs. We verified this disjointness empirically: running SWE-bench instances through our performance keyword filter and manually reviewing a random sample of 50 instances, all contained issue statements clearly indicating new behavior (validated by SWE-bench's `test_patch`).

### E.3. Identifying covering correctness tests

In this third stage, we retain a task instance only if the repository installs with all required dependencies, if the tests execute properly, and if we can identify unit tests that intersect the PR diff via line coverage. Installation is usually the most brittle step: a repo may install successfully yet fail at test time due to mismatched or missing dependencies. In practice, this requires manual curation—pinning versions and resolving transitive constraints—to map each instance to a working environment, beyond the constants provided by SWE-bench and SWE-Gym.

Once tests run cleanly, we execute the full test suite with coverage enabled and record, for each test, the lines of each source file that are executed. For every test file, we align this dynamic coverage with the lines, functions, classes, and modules modified in the PR to determine whether the test intersects the change. We keep a task instance only if at least one unit test intersects the original PR edit. Recall that SWE-bench selected PRs that introduced tests: since we intentionally select tests without test changes, this coverage step is required for us to identify guarding tests in the code repo.

Because our correctness check targets performance edits intended to be semantics-preserving, any check violation must appear on executions that traverse the modified regions. We therefore restrict the necessary correctness tests to those whose coverage intersects the PR diff (aggregated across lines, functions, classes, and modules). This change-focused selection is a conservative form of test-impact analysis: tests that never execute the modified code cannot surface regressions, yet they would inflate wall-clock time and noise in a benchmark setting. Limiting evaluation to intersecting tests preserves detection power for performance regressions, reduces spurious failures from unrelated tests, and yields stable, low-cost runs—making the benchmark practical for repeated use and community adoption.

**Static analysis pipeline and tooling.** Our coverage-guided test selection is implemented from five concrete components: (i) `tree-sitter` (via `tree_sitter_languages`) for the AST comparison in Stage II, ignoring comment and string nodes when checking whether the PR's pre- and post-edit ASTs differ meaningfully; (ii) `coverage.py` for per-test dynamic line coverage; (iii) Python's standard `ast` module together with `asttokens` for parsing source files and mapping AST nodes to line and column spans (used to identify imports, function and class definition headers, and type-hint regions that should be stripped from the executed-line set); (iv) `jedi` for cross-file symbol resolution, expanding each test's set of touched symbols via `Script.get_names(all_scopes=True, references=True)` followed by `goto()` to follow imports and assignments to their definitions; and (v) `unidiff` together with `intervaltree` for parsing the PR diff into per-hunk line sets and tracking per-file *(start_line, end_line)* intervals efficiently when aggregating reachable code across many symbols.

**End-to-end test selection.** Given a candidate PR, the test-selection pipeline proceeds as follows:

1. *Extract modified lines from the PR diff.* We parse the unified diff with `unidiff`. For each hunk, we record the post-edit line set: added lines directly, and (for chunks of removed lines) the surrounding post-edit lines bracketing the removal so that callers of the removed code are also represented.

2. *Collect per-test coverage.* We execute the repository's test suite under `coverage.py` and record, for each test, the (file, line) pairs that were executed.

3. *Symbol-level expansion.* For every test file, we walk all referenced symbols using `jedi` and follow `goto()` edges (up to a bounded depth) to discover transitively-referenced functions, classes, and modules in the source tree. We accumulate the resulting (file, line range) intervals in an `intervaltree`, giving each test a reachable-code footprint that complements its raw line coverage and recovers tests that exercise the modified code via wrapper functions or thin compiled shims.

4. *Strip false positives from coverage.* `coverage.py` marks function- and class-definition header lines as well as import statements as covered at import time, even when their bodies never run in a given test. We use `ast` and `asttokens` to detect and drop these lines, along with type-hint annotation spans, from each test's executed-line set.

5. *Intersect.* For each test, we intersect its filtered executed-line set (and its symbol-expansion intervals) with the PR's modified-line set. If the intersection is non-empty, we mark the test as a *guarding test* for that PR.

We retain only PRs that have at least one guarding test under this procedure, yielding the 1041 PRs reported in Section 2.1 prior to workload annotation and execution-based filtering. The full implementation is included in our released codebase.

**Distribution of covering tests per instance.** Because every test count reported in the paper refers to *covering* tests (i.e., tests whose dynamic line coverage or symbol-expansion intervals intersect the expert patch) rather than the host repository's total test suite, the per-instance distribution is itself an interesting statistic. Across the 498 tasks in SWE-FFICIENCY, the median number of covering tests is 7,723 per instance. Over 90% of tasks are guarded by more than 100 covering tests, and approximately 74% of tasks are guarded by more than 1,000 covering tests. This concentration is what makes the pass-to-pass correctness check meaningful even for small expert patches: a small diff can still be exercised by thousands of tests via wrapper functions, shared utilities, and module-level helpers that the symbol-expansion step in our pipeline recovers.

### E.4. Annotating performance workloads

Given performance-related candidate task instances for which we can easily check that edits maintain correctness of code, we need a way of also grading whether edits improve performance. We explored using an LM generated pipeline to generate workloads (see Appendix K), but found that manual annotation based on GitHub issue PR and issue metadata was a more effective strategy and yielded more realistic workloads (i.e. the same workloads that PR authors used as a baseline to implement their optimization edits). Thus, for each candidate task instance in this stage, we examine its linked GitHub pull request and issue info and generate a workload script which shows a performance delta: we double check these workloads in the next stage to verify the reproducibility and statistical significance of the performance improvements for benchmark inclusion.

Each workload consists of four items: (i) required imports (including `timeit` and `statistics` for computing runtime distributions); (ii) an optional `setup()` function for initializing any parts of the performance workload that should not be measured for runtime; (iii) a `workload()` function, which encapsulates the key functionality of interest to measure; and (iv) timing-specific code to run workloads multiple times to generate consistent runtime distributiosn for evaluation and analysis. See Appendix B.2 for examples of performance workloads and Appendix K for detailed workload creation instructions.

Notably, we find that automatically extracting workloads that show the performance delta from PR information is difficult. For example, `pandas` PRs #43274, #49596, #59608 each contain at least one (of many) codeblocks with a performance script from the PR author, showing the intended performance delta. However, note that each block uses a different format, timing mechanism, and method of executing programs, as shown in Figures 12, 13, 14. Unifying these consistently without hugely reducing dataset yield is non-trivial, as shown in other works like GSO (Shetty et al., 2025). We leave LM-pipeline based approaches to automate this extraction to future work.

**PR Performance Script Codeblock for `pandas-dev__pandas-43274`**

```
In [1]: from asv_bench.benchmarks.indexing import InsertColumns

In [2]: self = InsertColumns()

In [3]: self.setup()

In [4]: %timeit self.time_assign_with_setitem()
27.6 ms ± 68.1 µs per loop (mean ± std. dev. of 7 runs, 10 loops each) #master

In [4]: %timeit self.time_assign_with_setitem()
8.6 ms ± 6.43 µs per loop (mean ± std. dev. of 7 runs, 100 loops each) #PR
```

*Figure 12.* This PR uses existing `asv` benchmarks in the repository and measures performance improvement with the `timeit` command line entrypoint.

**PR Performance Script Codeblock for `pandas-dev__pandas-49596`**

```python
import pandas as pd
import pandas._testing as tm

vals = pd.Series(tm.rands_array(10, 10**6), dtype="string")
df = pd.DataFrame({"cat": vals.astype("category")})

%timeit df.groupby("cat").size()

1.21 s ± 4.71 ms per loop (mean ± std. dev. of 7 runs, 1 loop each)      <- main
15.1 ms ± 274 µs per loop (mean ± std. dev. of 7 runs, 100 loops each)  <- PR
```

*Figure 13.* This PR uses a bespoke workload (non `asv` benchmark and measures a specific functionality, again using `timeit` to measure speedup.

**PR Performance Script Codeblock for `pandas-dev__pandas-59608`**

```python
import pandas as pd
import pyarrow as pa
import pyarrow.csv as csv
import time

NUM_ROWS = 10000000
NUM_COLS = 20

# Example Multi-Index DataFrame
df = pd.DataFrame(
    {
        f"col_{col_idx}": range(col_idx * NUM_ROWS, (col_idx + 1) * NUM_ROWS)
        for col_idx in range(NUM_COLS)
    }
)
df = df.set_index(["col_0", "col_1"], drop=False)

# Timing Operation A
start_time = time.time()
df.to_csv("file_A.csv", index=False)
end_time = time.time()
print(f"Operation A time: {end_time - start_time} seconds")

# Timing Operation B
start_time = time.time()
df_reset = df.reset_index(drop=True)
df_reset.to_csv("file_B.csv", index=False)
end_time = time.time()
print(f"Operation B time: {end_time - start_time} seconds")
```

*Figure 14.* This PR uses both a bespoke workload and non-`timeit`, Python timing functionality to measure speedup.

To ensure a consistent and measurement environment for every PR, we pre-built Docker images with dependencies pre-installed. Each image encapsulated the repository at the specific `base` commit (i.e., immediately before the PR's changes were applied) along with all necessary dependencies to execute the project's code and test suites. Prior to commencing the full annotation task, the two author annotators calibrated their methodology and interpretation. They jointly annotated a set of five (5) example workloads, discussing discrepancies and establishing a consistent standard for what constituted a valid and representative workload. This calibration ensured alignment on the annotation process, including how to interpret PR descriptions, locate relevant code, and structure the final workload script. During annotation, the authors communicated regularly to ensure they stayed calibrated throughout. In total, workload annotation took around 200 author-hours for 1041 instances, 498 of which made it through final execution validation. Given how we initially scraped 100k PRs, this is a highly scalable pipeline to reduce the amount of manual annotation overhead

**Annotator Workload Template**

```python
# Import statements and one-time global work here
import timeit
import statistics
...

def setup():
    # Any one time work that needs to be done before every new run.
    pass

def workload():
    # Workload to be measured.
    pass

# Fill in number of repeats to get tight error bars
runtimes = timeit.repeat(workload, number=1, repeat=10, setup=setup)

print("Mean:", statistics.mean(runtimes))
print("Std Dev:", statistics.stdev(runtimes))
```

*Figure 15.* Example template provided to annotators to fill in and adapt with PR specific content.

For each of the 1041 PRs, the annotation task involved pulling the instance Docker image locally, then a deep analysis of the PR's description, code differential (`diff`), and any associated discussion threads, code blocks, or linked issues (including browsing GitHub and the state of the codebase at that commit). The goal was to identify or reconstruct the specific code path or usecase that the PR author intended to optimize, with high preference towards existing code blocks and PR author performance scripts. Annotators then scripted this workload into a standardized Python `def workload()` template (as shown in 15), which can then be executed against the repository. A rigorous two-step verification process was applied to every annotated workload before its inclusion in the benchmark:

1. **Peer Verification:** First, all workloads were verified by both annotators via discussion and approval. This side-by-side review process ensured the workload's logic was sound, it accurately captured the optimization described in the PR, and it was a faithful representation of the performance test (either explicitly provided by the PR author or inferred from the code changes).

2. **Execution Verification:** Second, each workload was programmatically executed to confirm its correctness and efficacy. We ran every workload script against two distinct versions of the code within its Docker container: (1) the `base` commit (pre-optimization) and (2) the post-patch state (post-optimization). A workload was only accepted if it executed successfully on both commits *and* demonstrated a statistically significant performance speedup on the `head` commit, thereby empirically validating the PR's performance claim.

Only workloads that passed both peer and execution verification were included in the final set of tasks for our benchmark. Additionally, all SWE-FFICIENCY tasks derive from *merged* pull requests—each PR was reviewed and accepted by repository maintainers, providing implicit validation that the optimization is worthwhile and the approach is sound. This validation standard is consistent with other major benchmarks: SWE-bench (Jimenez et al., 2024), GSO (Shetty et al., 2025), and Algotune (Press et al., 2025) all source tasks from merged PRs or curated problems without additional core-maintainer review beyond the merge itself.

### E.5. Execution-based filtering

We finally verify task instances by (i) executing and collecting their after-gold-edit test statuses and (ii) verifying that performance optimizations are statistically significant. Each annotated workload script contains a `workload()` function and a measurement harness to run a repeated number of iterations, generating a distribution of runtimes with both a mean and standard deviation (pre-edit as $\mu_{pre}$ and $\sigma_{pre}$ and $\mu_{post}$ and $\sigma_{post}$). We filter away any instances where $\mu_{pre} - \mu_{post} \leq 2\sigma_{post}$ (i.e. runtime speedup is larger than two post-edit runtime standard deviations). In this stage we also run correctness tests ten times to filter out any possible flaky tests, as those would cause our aggregated speedup ratio to be lower than the actual value.

# F. Techniques for Improving Performance Reproducibility

This section describes two implementation choices we use in our benchmark to reduce incidental variability in measured runtime and throughput: (i) prebuilding Docker containers so that environment resolution and installation never occur on the critical path of an evaluation run, and (ii) CPU pinning that separates container execution from Docker management daemons and assigns containers to non-overlapping groups of logical cores.

### F.1. Prebuilding instance docker images

We containerize each benchmark task and *prebuild* the corresponding Docker images prior to any timed evaluation (uploading it to a public Docker image registry). Thus, evaluation runs start from a fully built image; they do not perform package installation, environment resolution, or other setup work that would otherwise consume CPU cycles and introduce run-to-run variance. This design ensures that the CPU resources measured during evaluation are dedicated to the containerized program and harness rather than to container initialization. It also makes parallel execution more stable: because images are prepared ahead of time, concurrent workers do not contend for CPU due to on-the-fly dependency installation or environment setup. We provide these scripts in our code artifact release.

### F.2. Pinning Containers and Docker Daemon to CPU Cores

We implement a CPU-affinity policy that (a) assigns containers to disjoint groups of logical cores and (b) reserves a separate set of physical CPUs for Docker's background services. The policy proceeds as follows.

**Grouping logical cores.** Since each instance is evaluate on 4 vCPUs and 16GB of RAM, we first identify the logical-core (vCPU) topology and partition the available vCPUs into groups of four (4), with the constraint that *no two vCPUs in the same group share a physical core*. This grouping helps reproducibility because cache lines are generally isolated per physical core (and thus isolated between groups of 4 vCPUs), so execution within one group is more insulated from core-level contention within that group.

**Isolating Docker management.** We pin the Docker daemon and `containerd` to a dedicated set of *physical* CPUs (and their corresponding logical cores) that is disjoint from $\bigcup_i G_i$. As a result, container- and image-management activity (e.g., image downloading and setup) is confined to these reserved CPUs and cannot steal cycles from the cores executing benchmark containers. This separation allows us to parallelize workers across multiple groups $G_i$ without coupling their performance to background Docker activity.

**Memory limits and NUMA node binding.** In addition to CPU affinity, we restrict memory on a per-container basis, allowing each container to only consume 16GB and assign each container to use the NUMA (Non-Uniform Memory Access) memory node corresponding to the physical cores of the vCPUs that the container is assigned to. This bounds each worker's memory footprint and makes memory allocation more predictable and reduces cross-container interference due to host-level memory pressure, complementing the CPU isolation described above.

# G. Agent Harness Prompts and Details

This section describes the prompt and harness specific details used to generate our evaluation results in Section 4. More details can also be found in our attached code artifact.

### G.1. Code Optimization Task Prompts

The prompt provided to OPENHANDS and SWE-AGENT is provided below, asking an LM agent to optimize a specific workload given a repository, file utilties, and bash execution abilities in a containerized environment. Note that the agent is also given the commands for (1) rebuilding/reinstalling the repository and (2) the generic prefix command for running an arbitrary unit-test file.

---

**Code Optimization Task Prompt**

```
<uploaded_files>
{{working_dir}}
</uploaded_files>

I've uploaded a python code repository in the directory workspace_dir_name. Consider
↪  the following python workload showing a specific usage and measured performance
↪  of the repository:
<performance_workload>
```

---

```
{{workload}}
</performance_workload>
```

Can you help me implement the necessary changes to the repository so that the runtime
↪ of the `workload()` function is faster? Basic guidelines:

1. Your task is to make changes to non-test files in the /workspace directory to
↪ improve the performance of the code running in `workload()`. Please do not
↪ directly change the implementation of the `workload()` function to optimize
↪ things: I want you to focus on making the workload AS IS run faster by only
↪ editing the repository containing code that the `workload()` function calls.

2. Make changes while ensuring the repository is functionally equivalent to the
↪ original: your changes should not introduce new bugs or cause already-passing
↪ tests to begin failing after your changes. However, you do not need to worry
↪ about tests that already fail without any changes made. For relevant test files
↪ you find in the repository, you can run them via the bash command `{{test_cmd}}`
↪ <test_file>` to check for correctness. Note that running all the tests may take a
↪ long time, so you need to determine which tests are relevant to your changes.

3. Make sure the `workload()` function improves in performance after you make changes
↪ to the repository. The workload can potentially take some time to run, so please
↪ allow it to finish and be generous with setting your timeout parameter: for
↪ faster iteration, you should adjust the workload script to use fewer iterations.
↪ Before you complete your task, please make sure to check that the **original
↪ performance workload** and `workload()` function runs successfully and the
↪ performance is improved.

4. You may need to reinstall/rebuild the repo for your changes to take effect before
↪ testing if you made non-Python changes. Reinstalling may take a long time to run,
↪ so please be patient with running it and allow it to complete if possible. You
↪ can reinstall the repository by running the bash command `{{rebuild_cmd}}` in the
↪ workspace directory.

5. All the dependencies required to run the `workload()` function are already
↪ installed in the environment. You should not install or upgrade any dependencies.

Follow these steps to improve performance:

1. As a first step, explore the repository structure.

2. Create a Python script to reproduce the performance workload, execute it with
↪ python <workload_file>, and examine the printed output metrics.

3. Edit the source code of the repository to improve performance. Please do not
↪ change the contents of the `workload()` function itself, but focus on optimizing
↪ the code in the repository that the original `workload()` function uses.

4. If non-Python changes were made, rebuild the repo to make sure the changes take
↪ effect.

5. Rerun your script to confirm that performance has improved.

6. If necessary, identify any relevant test files in the repository related to your
↪ changes and verify that test statuses did not change after your modifications.

7. After each attempted change, please reflect on the changes attempted and the
↪ performance impact observed. If the performance did not improve, consider
↪ alternative approaches or optimizations.

8. Once you are satisfied, please use the finish command to complete your task.

*Table 6.* SWE-FFICIENCY results across several frontier models (higher is better; human-expert speedup ratio (SR) is 1.0×). SR is $pass@1$: each system submits a single patch per instance to be evaluated. SR is calculated by computing the speedup from the LM-generated edit, normalized by the speedup from the gold (human-written) patch, and aggregated across all tasks via harmonic mean.

| System | Speedup Ratio |
|---|---|
| Expert | 1.0× |
| CLAUDE 4.5 OPUS (OPENHANDS) | 0.225× |
| GPT-5 (OPENHANDS) | 0.157× |
| GPT-5.2 (OPENHANDS) | 0.148× |
| CLAUDE 4.5 SONNET (OPENHANDS) | 0.116× |
| GEMINI 3 FLASH (OPENHANDS) | 0.106× |
| GEMINI 3 PRO (OPENHANDS) | 0.102× |
| CLAUDE 4.1 OPUS (OPENHANDS) | 0.098× |
| QWEN3 CODER PLUS (OPENHANDS) | 0.068× |
| KIMI K2-0905 (OPENHANDS) | 0.054× |
| DEEPSEEK V3.1 (OPENHANDS) | 0.043× |
| GLM-4.6 (OPENHANDS) | 0.042× |
| GEMINI 2.5 FLASH (OPENHANDS) | 0.041× |
| GPT-5 MINI (OPENHANDS) | 0.039× |
| CLAUDE 3.7 SONNET (OPENHANDS) | 0.024× |
| GEMINI 2.5 PRO (OPENHANDS) | 0.031× |
| CLAUDE 3.7 SONNET (SWE-AGENT) | 0.041× |
| GPT 5 MINI (SWE-AGENT) | 0.026× |
| GEMINI 2.5 FLASH (SWE-AGENT) | 0.006× |

```
Please remember that you should not change the implementation of the `workload()`
↪   function. The performance improvement should solely come from editing the source
↪   files in the code repository.
```

### G.2. Details on Language Model Sampling Parameters

We elaborate on the evaluation settings discussed in 3. For all models, we perform the recommended greedy sampling within the OpenHands and SWE-agent harnesses. For GPT-5 MINI, we sample at a temperature of $t = 1$ (as mandated by the API as of August 2025) and at $t = 0$ for all other models. In the SWE-agent setting, we enforce a token spending limit of \$1, meaning that, in addition to the 100-turn action limit and time-limits specified, evaluation runs per-instance are stopped when (API/token-spending) cost is exceeded, and their patches as of that last action are immediately submitted (as is common practice with cost-limited SWE-agent runs on SWE-bench). We believe this lower-resource, cost-constrained setting is important from an *efficiency* standpoint of eventually yielding systems that can solve optimization tasks at reasonable dollar costs. In the OpenHands setting, our results only have the action count and time limits enforced. We also provide links to our forks of those agent harnesses for evaluation, which will also be merged upstream with corresponding harness libraries for community reproducibility.

## H. Additional Details on Main Evaluation Results

### H.1. Benchmark Performance Full Results

We provide full versions of evaluation results below for both OPENHANDS and SWE-AGENT below in Tables 6. Recall that for both harnesses, we set a 3 hour wall-clock time limit and a 100 turn interaction limit: in the SWE-AGENT case, we also limit LMs to a maximum cost of \$1.

### H.2. How does LM Performance Scale with Dataset Difficulty?

In addition to the bucketed trends shown in Figure 4, Figure 16 shows curves demonstrating how models perform on our dataset as we increasingly include tasks across the same three dimensions of (i) pre-edit workload duration, (ii) number of lines modified in the gold (expert) patch, and (iii) speedup factor achieved by the gold patch. We notice that as increased difficulty tasks are included, speedup ratio across the dataset decreases, further indicating that LMs are achieving easier wins on lower difficulty problems but struggling at higher difficulties.

*Table 7.* Distribution of patch outcomes by system. "Passes correctness tests" denotes functional correctness only (and not necessarily performance-optimal).

| System | Fails Tests (↓) | Passes Correctness Tests | | |
|---|---|---|---|---|
| | | Slower than Pre-edit (↓) | Faster than Pre-edit (↑) | Faster than Expert (↑) |
| CLAUDE 4.5 OPUS | 9% | 1% | 47% | 43% |
| GPT-5 | 18% | 4% | 32% | 46% |
| GPT-5.2 | 11% | 3% | 34% | 52% |
| CLAUDE 4.5 SONNET | 19% | 5% | 44% | 33% |
| GEMINI 3 FLASH | 25% | 5% | 33% | 37% |
| GEMINI 3 PRO | 14% | 3% | 36% | 47% |
| CLAUDE 4.1 OPUS | 15% | 4% | 43% | 38% |
| QWEN3 CODER PLUS | 18% | 13% | 44% | 25% |
| KIMI K2-0905 | 26% | 11% | 44% | 19% |
| DEEPSEEK V3.1 | 19% | 18% | 44% | 18% |
| GLM-4.6 | 33% | 13% | 38% | 16% |
| GEMINI 2.5 FLASH | 39% | 12% | 38% | 11% |
| GPT-5 MINI | 45% | 12% | 30% | 13% |
| CLAUDE 3.7 SONNET | 35% | 11% | 33% | 21% |
| GEMINI 2.5 PRO | 40% | 18% | 34% | 8% |
| CLAUDE 3.7 SONNET (SWE-AGENT) | 40% | 8% | 27% | 24% |
| GPT 5 MINI (SWE-AGENT) | 25% | 11% | 35% | 27% |
| GEMINI 2.5 FLASH (SWE-AGENT) | 44% | 12% | 35% | 8% |

### H.3. Examining More Expensive Reasoning Models: Comparing Gemini 2.5 Pro vs. Flash

We initially (as of September 19th 2025) could not run full benchmark results on full reasoning models like GPT-5, OPUS 4.1, and GEMINI 2.5 PRO *due to budget and runtime limitations*: runs would cost a significant amount and also take much longer per inference call (even with parallel requests) to reasonably complete in time.

Instead, we share results from a selected subset of 100 SWE-FFICIENCY problems, designated as SWE-FFICIENCY LITE. For this subset, we sample to be representative with respect to pre-edit workload runtime, gold patch speedup, and number of lines in gold patch from the distributions in Figure 3. Specifically, we construct a small, distribution-matched "lite" split by log-spacing each difficulty metric into bins and assigning instances to bins. We allocate a per-metric quota for a target size $N = 100$ in proportion to each bin's population and sample without replacement from those bins (using a fixed random seed). We take the union across metrics to cover diverse regions of each marginal distribution and top up any remaining slots by sampling from the unsampled pool with weights inversely proportional to the density of each instance's 3-way bin signature, which promotes rare metric combinations and preservess joint structure. If the union overshoots $N = 100$, we trim instances uniformly at random until we reach the desired amount.

In Table 8, we see that GEMINI 2.5 PRO performs similarly to its medium-compute counterpart GEMINI 2.5 FLASH on SWE-FFICIENCY LITE, while being more than $5\times$ as expensive dollar-wise and incurring an extra $2.74$ total hours of inference latency. This suggests that more-expensive state-of-the-art reasoning models also still heavily struggle on SWE-FFICIENCY and larger, agentic advances are needed to make models that reason more in-depth about repo-level performance and that can iterate in harnesses quickly.

*Table 8.* SWE-FFICIENCY LITE results between GEMINI 2.5 PRO and GEMINI 2.5 FLASH (higher is better; human-expert parity is $1.0\times$). "Passes tests" indicates passing functional correctness tests only. LM cost is total token spend (including prompt-caching). Inference latency is sum of total request latency over all requests.

| System | Speedup Ratio | Passes Tests | LM Cost | Inference Latency (hrs) |
|---|---|---|---|---|
| GEMINI 2.5 PRO (OPENHANDS) | 0.008× | 60% | $509.52 | 9.03 |
| GEMINI 2.5 FLASH (OPENHANDS) | 0.007× | 65% | $98.83 | 6.29 |

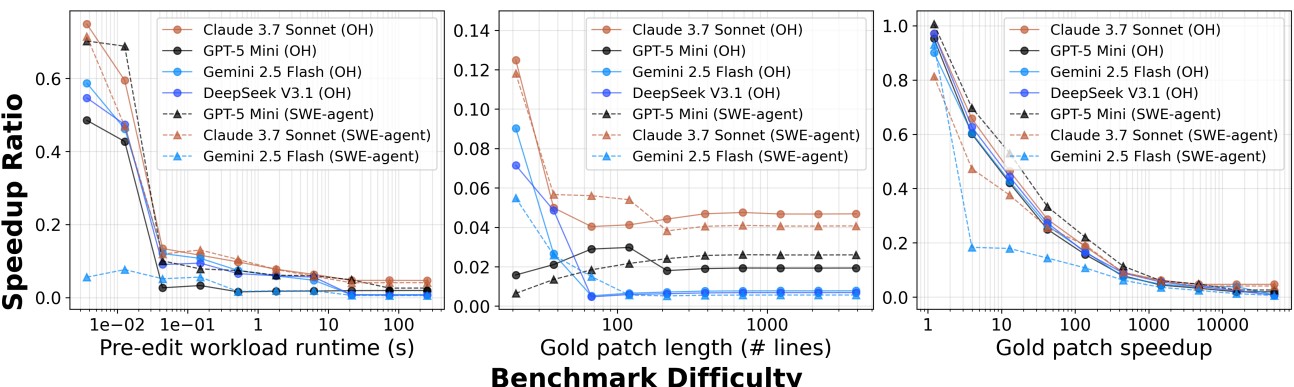

*Figure 16.* LMs achieve easier wins on lower difficulty problems, but struggle as higher difficulty tasks are included across multiple "definitions" of difficulty. For each difficulty measure and each measure upper bound $\tau$, we restrict to instances with difficulty measure $\leq \tau$ and report the resulting aggregate speedup ratio, generating our curves shown.

# I. Profiling-Based Attribution and Coverage Metrics

In this section, we elaborate on how we computed the profiling and function localization results shared in Section 4 and Figure 6, which show that LMs often miss out on expert-level speedup due to function level mislocalization.

**What is a function-level profiler?**   A function-level profiler instruments program execution to record, for every function invocation, (i) the *exclusive* or *self* time spent in the function body (excluding callees), often called *tottime*, (ii) the *inclusive* or *cumulative* time spent in the function and its transitive callees, often called *cumtime*, (iii) call counts, and (iv) the caller–callee relationships that induce a directed call graph. In our setting we profile a benchmark `workload` entry point before and after a code edit, obtaining two traces per actor (expert vs. LLM). We compute all metrics *only* on instances that pass correctness checks (e.g., unit or regression tests), so any measured speedup does not come at the expense of functional correctness.

## I.1. Data and Notation

Let $\mathcal{G}$ denote the set of functions observed by the profiler. We uniquely identify a function $g \in \mathcal{G}$ by its source file $f(g)$, line number $\ell(g)$, and name $n(g)$. For an actor $A \in \{\text{Expert}, \text{LLM}\}$ and a profiling phase $p \in \{\text{pre}, \text{post}\}$, let

$$\tau_{A,p}(g) \in \mathbb{R}_{\geq 0} \quad \text{and} \quad T_{A,p}(g) \in \mathbb{R}_{\geq 0}$$

denote the exclusive (*tottime*) and inclusive (*cumtime*) runtime attributed to $g$, respectively. We define per-function improvements (positive means faster) as

$$\Delta_A^{\text{tot}}(g) := \tau_{A,\text{pre}}(g) - \tau_{A,\text{post}}(g), \qquad \Delta_A^{\text{cum}}(g) := T_{A,\text{pre}}(g) - T_{A,\text{post}}(g).$$

Let $W$ denote the top-level entry point (`workload`); its end-to-end improvement for actor $A$ is

$$\delta_A^W := T_{A,\text{pre}}(W) - T_{A,\text{post}}(W).$$

We additionally report whole-trace speedups normalized by pre-edit workload time,

$$\text{Speedup}_A^W := \frac{\delta_A^W}{T_{A,\text{pre}}(W)}, \qquad \text{Speedup}_A^{\text{tot}} := \frac{\sum_g \tau_{A,\text{pre}}(g) - \sum_g \tau_{A,\text{post}}(g)}{T_{A,\text{pre}}(W)}.$$

**Call graph and depths.**   We first need to isolate the call graph (and function runtimes) that are strictly attributed to the `workload` function in each performance workload (and disregard function runtimes from any other source, like the `setup` function). From the *pre*-edit profile we build a directed call graph $\mathcal{C} = (\mathcal{G}, E)$ whose edges point from caller to callee. We define the *workload depth* $d(g)$ as the minimum caller distance from any node named `workload` to $g$ in $\mathcal{C}$; nodes not reachable from `workload` have undefined depth. Depths are used only for selection and the diagnostic depth metric in Appendix I.5.

**Patch-based file filters.**   We parse unified diffs to extract modified files for each actor and restrict candidate functions to those files. This ensures we attribute improvements to edited regions and reduces noise from unrelated code.

## I.2. Selecting Core Function-Level Improvements Without Double Counting

Naively summing $\Delta_A^{\text{cum}}$ over functions double-counts speedups because a caller's inclusive time subsumes callee improvements. We therefore select a *deepest non-overlapping* set of improved functions for each actor via a depth-aware greedy procedure.

**Thresholding and candidates.** We set a per-instance absolute threshold

$$\theta := \max\big(\theta_{\text{sec}},\ \theta_{\text{frac}} \cdot \delta_{\text{Expert}}^W\big),$$

where $\theta_{\text{sec}}$ is an absolute time floor (seconds) and $\theta_{\text{frac}}$ is a fraction of the expert's end-to-end improvement (default $0.02$). Candidates functions are defined below where $W$ is the node corresponding to the `workload` function entry point (see B.2 for a workload example):

$$\mathcal{C}_A := \left\{ g \in \mathcal{G} : \begin{array}{l} \Delta_A^{\text{cum}}(g) > 0, \Delta_A^{\text{cum}}(g) \geq \theta, \\[4pt] g \text{ is reachable from } W, \\[4pt] \text{and} f(g) \text{ was edited by } A \end{array} \right\}.$$

Let $S_A^+ := \sum_{g \in \mathcal{C}_A} \Delta_A^{\text{cum}}(g)$ denote the total positive mass among candidates.

**Greedy deepest-first selection.** We sort candidates by (i) larger depth $d(g)$ first, (ii) larger share $\Delta_A^{\text{cum}}(g) / \max(T_{A,\text{pre}}(W), \varepsilon)$, then (iii) larger $\Delta_A^{\text{cum}}(g)$ (ties broken arbitrarily), and greedily build a set $E_A \subseteq \mathcal{C}_A$ such that no selected function is an *ancestor* (caller, transitively) of another selected function in the pre-edit call graph. Intuitively, we select a set of functions closest to the scope of the speedup (i.e. the function scope where the speedup has the most significant percentage improvement over that time scope). For example, a speedup could occur in function $C$ but is called by $B$, which is then called by $A$: $C$ would show the largest percentage speedup relative to the total amount of time spent in that function, since $B$ and $A$ have other runtime overhead that was not optimized. We stop when the accumulated mass reaches a configurable cap $\rho \in (0, 1]$:

$$\sum_{g \in E_A} \Delta_A^{\text{cum}}(g) \ \geq \ \rho \cdot S_A^+ \quad (\text{default } \rho = 1).$$

If the procedure would select nothing, we include the single best candidate. This yields our set of functions $E_{\text{Expert}}$ and $E_{\text{LLM}}$.

## I.3. Expert-Relative Coverage (ERC) and Loss Decomposition

We measure how well the LLM's edits localize to the same *places of improvement* as the expert. Let $\Phi(S) := \{ f(g) : g \in S \}$ map a set of functions to its set of files. To define the expert's *attribution mass* we (i) keep only functions above threshold and (ii) restrict to files the expert actually optimized (to guard against spurious activity in unrelated files):

$$\mathcal{M}_{\text{Expert}} := \big\{ g \in \mathcal{G} : \Delta_{\text{Expert}}^{\text{cum}}(g) \geq \theta,\ f(g) \in \Phi(E_{\text{Expert}}) \big\}.$$

Define per-function expert mass $s_{\text{exp}}(g) := \Delta_{\text{Expert}}^{\text{cum}}(g)$ for $g \in \mathcal{M}_{\text{Expert}}$ and $0$ otherwise, and let $S_{\text{exp}} := \sum_g s_{\text{exp}}(g)$.

**Coverage at file and function granularity.** We compute ERC at two granularities:

$$\text{ERC}_{\text{file}} := \frac{\sum_{f \in \Phi(E_{\text{LLM}})} \sum_{g \in \mathcal{M}_{\text{Expert}} : f(g) = f} s_{\text{exp}}(g)}{S_{\text{exp}}},$$

$$\text{ERC}_{\text{func}} := \frac{\sum_{g \in E_{\text{LLM}}} s_{\text{exp}}(g)}{S_{\text{exp}}}.$$

Intuitively, $\text{ERC}_{\text{file}}$ asks "did the LLM edit the *right files*?" while $\text{ERC}_{\text{func}}$ asks "did it optimize the *right functions* within those files?"

**Loss decomposition.** We decompose the portion of expert mass *not* captured at the function level into two orthogonal failure modes:

$$\text{WrongFileLoss} := 1 - \text{ERC}_{\text{file}},$$

$$\text{InFileLoss} := \max\big\{0,\ \text{ERC}_{\text{file}} - \text{ERC}_{\text{func}}\big\}.$$

This yields a tight partition of expert mass:

$$\text{WrongFileLoss} + \text{InFileLoss} + \text{ERC}_{\text{func}} \;=\; 1,$$

where *WrongFileLoss* captures file-selection mistakes and *InFileLoss* captures localization mistakes *within* the right files (e.g., editing non-bottleneck functions).

### I.4. Edited-File Overlap

We also report the Jaccard similarity of edited files between actors:

$$\text{Jaccard} \;:=\; \frac{\left|\Phi(E_{\text{Expert}}) \cap \Phi(E_{\text{LLM}})\right|}{\left|\Phi(E_{\text{Expert}}) \cup \Phi(E_{\text{LLM}})\right|}.$$

When the union is empty (neither actor meets the selection threshold), the quantity is undefined and we omit it.

### I.5. Depth of Optimization from the Workload

As a diagnostic we compute a depth-of-optimization statistic with respect to the pre-edit call graph and the workload root. Let $[x]_+ := \max\{x, 0\}$ and define weights $w_A(g) := [\Delta_A^{\text{tot}}(g)]_+$ for $g \in E_A$. The *weighted average workload depth* and its coverage are

$$\overline{d}_A \;:=\; \frac{\sum_{g \in E_A \cap \text{reach}(W)} w_A(g)\, d(g)}{\sum_{g \in E_A \cap \text{reach}(W)} w_A(g)}, \qquad \text{ReachShare}_A \;:=\; \frac{\sum_{g \in E_A \cap \text{reach}(W)} w_A(g)}{\sum_{g \in E_A} w_A(g)}.$$

$\overline{d}_A$ reflects whether improvements concentrate near the entry point or deep in the call tree; $\text{ReachShare}_A$ indicates how much of the selected mass is reachable from the workload (should be close to 1 in well-instrumented runs). We use $\theta_{\text{sec}} = 0$, $\theta_{\text{frac}} = 0.02$ (i.e., a per-function floor at 2% of the expert's end-to-end gain $\delta_{\text{Expert}}^W$ to disregard speedups that are due to measurement noise), and cap $\rho = 1.0$, and we restrict candidate functions to edited files for each actor. The call graph used for depths and ancestry tests is always taken from the pre-edit trace to avoid post-edit structural confounds. Note that we only compute these statistics over instances that have passed functional correctness tests.

**Why $\Delta^{\text{cum}}$ for selection and $\Delta^{\text{tot}}$ for depth weights?** We select by $\Delta^{\text{cum}}$ to capture inclusive speedups (including callee effects) while the depth statistic weights by $\Delta^{\text{tot}}$ to avoid double-counting along a chain. The deepest-first greedy constraint further prevents attributing the same improvement to both a caller and its callee.

**Interpretation.** High $\text{ERC}_{\text{file}}$ with low $\text{ERC}_{\text{func}}$ indicates that the model navigated to the right files but failed to touch the expert-optimized functions (*within-file localization gap*). Low $\text{ERC}_{\text{file}}$ indicates a file-selection gap. Because $S_{\text{exp}}$ is defined over expert-selected files above threshold, the metrics focus on *where* expert improvements actually occurred, rather than on unrelated noisy regions.

### I.6. Full Speedup Attribution Results

| System | ERC$_{\text{file}}$ | ERC$_{\text{func}}$ | WrongFileLoss | InFileLoss | Jaccard (files) |
|---|---|---|---|---|---|
| CLAUDE 3.7 SONNET (SWE-AGENT) | 0.630 | 0.298 | 0.370 | 0.332 | 0.636 |
| CLAUDE 3.7 SONNET (OPENHANDS) | 0.611 | 0.314 | 0.389 | 0.297 | 0.604 |
| GPT-5 MINI (OPENHANDS) | 0.551 | 0.278 | 0.449 | 0.274 | 0.559 |
| GEMINI 2.5 FLASH (OPENHANDS) | 0.549 | 0.265 | 0.451 | 0.283 | 0.556 |
| DEEPSEEK V3.1 (OPENHANDS) | 0.519 | 0.246 | 0.481 | 0.273 | 0.531 |

*Table 9.* Expert-Relative Coverage (ERC) and related losses. Means computed over instances that pass correctness and have speedup $\geq 1$. File-overlap Jaccard is averaged over instances where it is defined.

We compute results only over LM patches that passed correctness and achieve a speedup from the pre-edit runtime (but not necessarily faster than the expert). Across systems, the ERC and loss metrics in Table 9 show a consistent pattern: *WrongFileLoss* is roughly $37\%-45\%$ across models (mean $\approx 41.5\%$), while *InFileLoss* is roughly $27\%-33\%$ (mean $\approx 29.7\%$). Taken together, this implies that models choose the wrong function (either by editing the wrong file or the wrong function within the right file) about $X+Y \approx 69\%-73\%$ of the time (mean $\approx 71.2\%$), consistent with $\text{ERC}_{\text{func}} \approx 0.26-0.31$. For context on number of instances analyzed per system and how deep optimizations occur in the call tree, Table 10 reports $n$ per system and the weighted average depth from the `workload` root (experts typically operate at call-stack depth $4.5-4.9$, LLMs at $3.8-4.2$).

| System | Number Correct Instances | Depth$_{exp}$ | Depth$_{llm}$ |
|---|---|---|---|
| CLAUDE 3.7 SONNET (SWE-AGENT) | 196 | 4.85 | 4.20 |
| CLAUDE 3.7 SONNET (OPENHANDS) | 252 | 4.61 | 4.18 |
| GPT-5 MINI (OPENHANDS) | 193 | 4.59 | 4.13 |
| GEMINI 2.5 FLASH (OPENHANDS) | 211 | 4.51 | 3.83 |
| DEEPSEEK V3.1 (OPENHANDS) | 246 | 4.89 | 4.28 |

*Table 10.* Dataset size ($n$) and weighted average optimization depth from the `workload` entry point in the pre-edit call graph (expert vs. LLM).

## J. Comparison of LM Generated Edits Versus Experts

We provide the raw diffs with in-line comments from Figures 8 and 9 below in Figures 17 and 18. We also include an additional diff for 19 comparison. In the main text, we removed comments and surrounding lines to focus only on the lines changed between expert and LM generated diff.

## K. Synthetically Generating Performance Workloads

We provide more details on our investigation on whether LMs can capably generate performance workloads, as discussed in the end of Section 4.2.

**Rationale.** We generate LLM-based workloads to mirror our human curation process and test a key hypothesis: when the *gold patch is held fixed*, workloads curated by expert annotators (our pipeline) expose larger, statistically reliable performance deltas than workloads produced by an LLM from the same evidence.

**Inputs per instance.** For each SWE-FFICIENCY benchmark instance we use:

- The unified diff of the expert (gold) patch,

- The pre-edit source files corresponding to paths touched in the diff, repository and commit identifiers.

**Prompt construction.** We parse the diff headers to identify touched files and fetch their pre-edit contents at the base commit. The model is given:

1. The full patch (pre/post diff), and

2. The concatenated pre-edit files for those paths

**Instruction parity with human annotators.** We prompt the LLM (GEMINI 2.5 FLASH) with similar instructions as we used ourselves for workload annotation: produce a *self-contained Python workload script* with `setup()` (realistic inputs; seeded randomness) and `workload()` (representative, non-trivial call path), executed via `timeit.repeat(...)`, and printing exactly two lines—mean and standard deviation. This parity isolates the effect of workload design quality, not interface differences. We provide the instruction prompt below.

---

**Synthetic Workload Generation Prompt**

```
You are a performance testing expert. You will be provided a code edit as a git diff
↪  and the pre-edit source files. You need to generate a **self-contained Python
↪  performance workload script** that measures perfomance of code paths or APIs
↪  changed in the diff.

Guidelines for the workload script contents.

- Use a `setup()` function to prepare any realistic, non-trivial data or environment
↪  needed for the test.
  - Data must be representative of real-world usage (avoid trivial arrays or easily
  ↪  optimizable patterns).
  - Prefer real datasets or realistic synthetic data with reproducibility (set a
  ↪  random seed).
  - All expensive or one-time setup (e.g., file download, preprocessing) must be in
  ↪  `setup()`, not in the workload.
```

---

- Use a `workload()` function to run the actual operation(s) being timed.
  - The workload should reflect a **representative and challenging real-world use
  ↪  case** of the API or library under test.
  - Avoid corner cases that could be trivially optimized.
  - Inputs should be varied enough to prevent caching or constant-folding from
  ↪  affecting results.

- Run the benchmark using `timeit.repeat(workload, number=..., repeat=...,
↪  setup=setup)`.
  - `number` should match a realistic single-run execution count (do not batch
  ↪  multiple runs for cumulative timing).
  - `repeat` should be high enough to gather stable statistics.

- Print the mean and standard deviation of the last set of runtimes using
↪  `statistics.mean()` and `statistics.stdev()`.
  - Output should be clear and ready for performance comparison.

- The output must be a **complete Python script** containing only:
  1. import statements
  2. `setup()` function
  3. `workload()` function
  4. the `timeit.repeat()` call
  5. mean/stddev printing

The script should only print two lines at the end: the mean of measured runtimes and
↪  the standard deviation of runtimes.

Example workload to follow (please strictly follow this format of imports, setup
↪  function, workload function, timeit call, and print statements). In particular,
↪  make sure the mean and standard deviation print statements are exactly as shown
↪  below.

```python
import timeit
import statistics
import numpy as np

def setup():
    global arr
    np.random.seed(42)
    arr = np.random.rand(5000, 5000)

def workload():
    global arr
    _ = arr @ arr.T

runtimes = timeit.repeat(workload, number=1, repeat=10, setup=setup)

print("Mean:", statistics.mean(runtimes))
print("Std Dev:", statistics.stdev(runtimes))
```

Here's a commit and it's information that does some optimization in the {repo_name}
↪  repository that might be relevant to writing the test:
## Commit Diff:
```

{commit_diff}
```

## Pre-edit source files:
{pre_edit_code}

**Evaluation (holding the patch fixed).** For each instance $i$, we evaluate both workloads—LLM-generated and manually annotated—against the same code states:

1. **Base**: Repository at the pre-gold patch commit.

2. **Patched**: Repository with the gold patch applied.

We then compare the magnitude of improvements (speedups) between the LM-generated (GEMINI 2.5 FLASH) workload and the manual workload for the same instance and compute if values are statistically significant and if the LM generated workload outperforms the annotated one. This ablation directly measures whether an LLM, given the same diff and file context and the same instructions as experts, can generate workloads that surface the patch's performance gains as reliably and strongly as manual curation. We see that, in 76% of cases, our manual annotations show a larger performance delta than the LM generated workloads on the expert patch, with 47% of workloads showing a non-significant performance delta at all.

**Caveats on the LM-workload-generation comparison.** Our pilot study is best interpreted as a *lower bound* on what LM-synthesized workloads can achieve, for two reasons. First, we used GEMINI 2.5 FLASH, a medium-capability model—frontier reasoning models (e.g., CLAUDE 4.5 OPUS, GPT-5) may produce stronger workloads, and we expect the manual-vs-LM gap to narrow as model capability improves. Second, the LM was given only the unified diff and pre-edit source files, whereas the manual annotators additionally had access to the PR description and discussion threads, which often contain the author's intended demo script and target operating point. The asymmetry was intentional for this ablation—we wanted to test whether code-only context is sufficient—but a fairer head-to-head would supply the LM with the same PR metadata. Manual annotation remains the most reliable choice given existing tooling, and we frame *LMs-in-the-loop workload generation* (frontier model + PR-description context + verification harness) as a concrete future-work direction enabled by SWE-FFICIENCY's curation infrastructure.

# L. Preventing Model Reward Hacking

In this section, we outline two techniques implemented in our evaluation harness to prevent LM reward hacking behavior. The first, *stack-frame-based* reward hacking covers when models try to determine their caller (i.e. whether they are being called in a performance runtime environment), while the second covers *run-to-run caching*, which is when models try to cache computations across evaluation runs to improve performance (but technically cheating, as we'd like to evaluate specific workloads under non-cached conditions).

### L.1. Preventing Stack-Frame–Based Reward Hacking

**Overview.** We observed that some LLM-generated patches only speed up workloads when they detect they are being timed (i.e. under `timeit` or from being called from a function with name `workload`, as is done in our benchmark). The common mechanism is Python stack introspection (e.g., `inspect.currentframe()`, `traceback.extract_stack()`, `sys._getframe()`, or reaching frame objects via `f_back`/`tb_frame`). These edits can short-circuit code paths or memoize based on caller identity, inflating measured speedups without actually improving the underlying algorithm.

In general, code changes (and in-particular performance improving edits) should *never* require caller identity information to improve performance. Other types of changes, such as tuning specifically based on input attributes (like size or data-type), are actually quite common in performance optimization PRs: we intentionally permit these changes in SWE-FFICIENCY. We verify that none of the expert PRs and gold patches use stackframe information: the one exception is `pandas-dev__pandas-45247`, which optimizes `find_stack_level` in pandas, a utility to more readably show exception stackframes (and where the expert patch uses these utilities).

**Goal.** We want to flag *newly introduced* stack-introspection logic in a submitted patch while tolerating any pre-existing usage in the codebase (many mature projects legitimately use `inspect`/`traceback` during import/configuration) as well as making sure that expert/gold patches are not falsely flagged.

**Implementation.** Our checker takes a unified diff as input and analyzes only the *post-image* of files touched by that diff. It reports an error *only* if the diff *adds* lines that contain stack-introspection primitives. Existing occurrences are ignored by design.

1. **Patch-scope extraction.** We parse the unified diff to recover, for each touched file, the set of line numbers that are newly added on the "+" side. This yields a map `added_lines : path ↦ {new line numbers}`. We also track which files are brand-new in the patch and the set of all post-image paths seen in +++ headers.

2. **Standalone-new filtering.** Brand-new files (introduced by the patch) that are *not imported* by any other file touched in the patch are treated as "standalone" and excluded from the check. This avoids flagging ad hoc scripts (e.g., local reproducer/benchmark drivers) that do not affect the library under test, as often LM systems might produce ad hoc scripts like these for debugging and introspection in a scratchpad form (which may use some stackframe inspection). This lets us ignore scratch-pad like files introduced by patches, while still checking newly-created files that are imported and used in the repository. We determine "referencedness" by building a lightweight import graph among touched files:

   (a) For each patch edited file, collect its imported module names via AST (both `import m` and `from m import x`).
   (b) For each brand-new file, derive candidate module names from its path (e.g., `foo/bar/baz.py` → {`baz`, `bar.baz`, `foo.bar.baz`}).
   (c) Mark the new file as referenced if any other touched file imports one of its candidates (exact or dotted-suffix match).

3. **Post-edit AST scan with alias resolution.** For each edited file in added_lines, we read the *post-edit* source file. We parse each post-edit file source with `ast` and walk the tree once, collecting "findings" whenever the code contains introspection-like constructs. This scan is robust to renaming via an an AST-based import resolve, described below

   - **Imports:** record module aliases (e.g., `import inspect as ins`) and function aliases (e.g., `from inspect import currentframe as cf`).
   - **Direct calls:** resolve callee to (module, attribute) pair and match against a denylist, which includes the following list:
     - `inspect.{currentframe, stack,getouterframes,getinnerframes,trace, getframeinfo,getsource,getsourcefile}`
     - `traceback.{extract_stack,format_stack,print_stack,walk_stack}`;
     - `sys.{_getframe,settrace,setprofile}`
     - `gc.{get_referrers,get_objects}`
   - **Dynamic imports:** Detect dynamic imports such as `__import__('inspect')` and `importlib.import_module('inspect')`.
   - **Frame-object attributes:** flag attribute reads commonly used to climb or expose frames (`.f_back`, `.tb_frame`, `.gi_frame`, `.cr_frame`, `.ag_frame`), regardless of receiver type (conservative heuristic).

4. **Added-line projection.** After AST scanning, we *project* identified usages (and line number occurrences) onto the added-line set for that file and retain only those whose source line number is in added_lines[*path*]. This makes the check purely *diff-relative*: modifications that reuse pre-existing introspection do not fail the patch (as those usages are valid).

5. **Reporting.** If any filtered findings remain (after standalone-new filtering and pragma suppression), a patch fails the check and the LM generated patch is considered incorrect and fails correctness in our evaluation.

We use AST parsing instead of regex/simple grep since simple regexes miss aliased imports and produce many false positives/negatives on strings or comments. Our AST pass is cheap, robust, and semantically aware: it resolves `inspect` aliases, identifies function calls regardless of whitespace/nesting, and recognizes dynamic imports. We verify that this mechanism (i) fails our previously identified LM generated patches that exploit stackframe info and (ii) passes our gold, expert edits (i.e. does not flag any false positives).

### L.2. Preventing Run-To-Run Cache Reward Hacking

**Overview.** Repeated measurements within a single Python process allow module- and process–local state to leak across runs (e.g. module-level dictionaries, `@lru_cache`, global arrays, ad hoc memo tables). Such state can make later iterations appear faster without changing the true cost of the underlying algorithm. To make per-iteration results robust, we isolate runs so that no Python-level caches (or mutated globals) can persist between repetitions.

**Implementation.** We transform our annotated `workload` scripts into an equivalent program that executes each timing repetition in a *fresh child process*. Concretely, we (i) parse the original script with `ast`, (ii) preserve its logic and output formatting, and (iii) replace the in-process `timeit.repeat` loop with a small harness built on `multiprocessing` using the *spawn* start method. The `fork` policy starts a brand-new copied process for every repetition, guaranteeing a

module namespace and empty caches equivalent to the original parent run. This implementation allows us to provide simple scripts at inference time in problem statements to LM agents, while also being able to use those scripts as inputs to yield memory-isolated runtime scripts.

1. **Locate the timing site.** We walk the AST to find the assignment to `runtimes = timeit.repeat(...)` (or an equivalent import form), then extract the *workload* callable, optional *setup* callable, and the numeric `number`/`repeat` parameters. We also record if the script later slices the results (e.g., `runtimes[-10000:]`) so that summary statistics are computed over the same view.

2. **Preserve surrounding code.** The transformer keeps all top-level declarations and statements *except* the original `timeit.repeat` assignment and the immediately following summary prints. Any statements that originally lived between those two points are preserved and either (i) executed after the isolated timing (if they are harmless post-processing) or (ii) moved into a guarded `finally` block if they look like teardown of temporary files/directories (simple heuristic over `os`/`shutil` calls).

3. **Fork-per-run harness.** We synthesize a minimal harness:

   - a child-side function that constructs a `timeit.Timer(workload, setup=setup)` and calls `Timer.timeit(number)`,
   - a top-level *picklable* target that runs the child once and returns the duration through a `multiprocessing.Queue`.
   - a driver `_run_isolated(number, repeat, start_method="fork")` that loops `repeat` times: for each iteration it creates a new process/context, executes the child, checks the exit code, and appends the reported duration.

   We deliberately use **fork** so the child interpreter starts from the same memory state as the parent (with copy-on-write) such that any edits to parent memory objects, such as module level Python caches (including `@lru_cache` and ad hoc dictionaries) cannot carry over between repetitions.

4. **Result and summary fidelity.** After the harness returns the list of durations, we reconstruct the original slicing intent (if any) into a `runtimes_view` and compute `Mean` and `Std Dev` exactly as in the input script. Any non-teardown statements that originally ran between the timing and the summary are executed afterward to preserve observable side effects.

Writing the transformation allows us to guarantee each repetition runs in a brand-new interpreter process; thus module-level state, Python memo tables, and global variables cannot influence subsequent repetitions. Import-time effects reoccur per iteration, making "first-run vs. warmed-run" behavior explicit in the measurement. Random number generators also begin from the child's fresh state unless the user seeds them in `setup`, in which case seeding is applied identically per run. By enforcing *fork-per-run*, the rewritten benchmarks are robust to module-level caching and other intra-process artifacts. The transformation preserves user-visible behavior (including result slicing and post-processing) while ensuring that any speedups reflect genuine algorithmic improvements rather than residual state from previous iterations.

---

**Original Raw Workload for `pandas-dev__pandas-56508`**

```python
import timeit
import statistics

import pandas as pd
import numpy as np

np.random.seed(0)

N = 100_000
data = np.arange(N)
arr = pd.array(np.where(np.random.rand(N) > 0.1, data, np.nan), dtype="Int32")

def workload():
    arr._hash_pandas_object(encoding='utf-8', hash_key="1000000000000000",
    ↪  categorize=False)
```

---

```
runtimes = timeit.repeat(workload, number=5, repeat=1000)

# Print runtime mean and std deviation.
print("Mean:", statistics.mean(runtimes))
print("Std Dev:", statistics.stdev(runtimes))
```

**Transformed and Memory Isolated Workload for `pandas-dev__pandas-56508`**

```
import timeit
import statistics
import pandas as pd
import numpy as np
np.random.seed(0)
N = 100000
data = np.arange(N)
arr = pd.array(np.where(np.random.rand(N) > 0.1, data, np.nan), dtype='Int32')
def workload():
    arr._hash_pandas_object(encoding='utf-8', hash_key='1000000000000000',
    ↪  categorize=False)

# ---- AUTO-GENERATED ISOLATION HARNESS (timeit-in-child, fork-safe) ----
import statistics as _statistics
import multiprocessing as _mp
import timeit as _timeit

def _child_once(number: int):
    setup_fn = (lambda: None)
    t = _timeit.Timer(workload, setup=setup_fn)
    return t.timeit(number)

# TOP-LEVEL target: picklable under 'fork'
def _child_target(q, number):
    try:
        dur = _child_once(number)
        q.put(dur)
    except BaseException:
        import traceback
        traceback.print_exc()
        q.put(None)

def _run_isolated(number: int, repeat: int, start_method: str = "fork"):
    ctx = _mp.get_context(start_method)
    results = []
    for _ in range(repeat):
        q = ctx.Queue()
        p = ctx.Process(target=_child_target, args=(q, number))
        p.start()
        dur = q.get()
        p.join()
        if dur is None or p.exitcode != 0:
            raise RuntimeError("Child run failed--see traceback above.")
        results.append(dur)
    return results

if __name__ == "__main__":
    _number = 5
    _repeat = 1000
    runtimes = _run_isolated(_number, _repeat, start_method="fork")
    runtimes_view = runtimes
```

```
    print("Mean:", _statistics.mean(runtimes_view))
    print("Std Dev:", _statistics.stdev(runtimes_view) if len(runtimes_view) > 1 else
    ↪  0.0)
```

## M. Full Example Diffs of LM Generated Edits Versus Experts

We provide the raw diffs with in-line comments from Figures 8 and 9 below in Figures 17 and 18. We also include an additional diff for 19 comparison. In the main text, we removed comments and surrounding lines to focus only on the lines changed between expert and LM generated diff.

## N. Speedup Ratio Stability Analysis and Error Bars

We also conduct a small study analyzing the variability and stability in our aggregate metric, Speedup Ratio (SR), since individual workload performance measurement can have minor variability. However, we find that SR does not vary significantly between independent runs. While raw runtime may exhibit micro-variance, the expert-grounded speedup normalization used in SR acts as an effective noise filter. To demonstrate this, we performed three independent evaluation runs of the patches generated by three selected LM agents as well as the expert edit, computing the SR for each run. As shown in Table 11, the variance in the final SR score is negligible ($< 0.5\%$), confirming that measurement noise does not affect the relative ranking or assessment of agent capabilities. This observation aligns with related work finding that conversion to normalized or percentage-based speedup metrics inherently reduces variability in final scores (Liu et al., 2024).

*Table 11.* Stability of Speedup Ratio (SR) across three independent runs. The low spread confirms that SR is robust to microbenchmark runtime variance.

| System | SR Run 1 | SR Run 2 | SR Run 3 | SR Spread |
|---|---|---|---|---|
| Expert | $1.000\times$ | $0.9986\times$ | $1.0026\times$ | $0.0040\times$ |
| Claude 3.7 Sonnet (OpenHands) | $0.04761\times$ | $0.04796\times$ | $0.04779\times$ | $0.00035\times$ |
| GPT-5 Mini (OpenHands) | $0.02115\times$ | $0.02115\times$ | $0.02128\times$ | $0.00013\times$ |
| Gemini 2.5 Flash (OpenHands) | $0.007941\times$ | $0.007947\times$ | $0.007935\times$ | $0.00012\times$ |

```
--- a/pandas/core/arrays/arrow/array.py
+++ b/pandas/core/arrays/arrow/array.py
@@ -406,8 +406,14 @@ def _cmp_method(self, other, op):
                f"{op.__name__} not implemented for {type(other)
      }"
            )

-        result = result.to_numpy()
-        return BooleanArray._from_sequence(result)
+        if result.null_count > 0:
+            # GH50524: avoid conversion to object for better perf
+            values = pc.fill_null(result, False).to_numpy()
+            mask = result.is_null().to_numpy()
+        else:
+            values = result.to_numpy()
+            mask = np.zeros(len(values), dtype=np.bool_)
+        return BooleanArray(values, mask)

    def _evaluate_op_method(self, other, op, arrow_funcs):
        pc_func = arrow_funcs[op.__name__]
```

```
--- a/pandas/core/arrays/arrow/array.py
+++ b/pandas/core/arrays/arrow/array.py
@@ -406,8 +406,16 @@ class ArrowExtensionArray(OpsMixin,
    ExtensionArray):
                f"{op.__name__} not implemented for {type(other)
      }"
            )

-        result = result.to_numpy()
-        return BooleanArray._from_sequence(result)
+        # Fast path: if there are no nulls, we can avoid the
    expensive BooleanArray creation
+        if result.null_count == 0:
+            # Ensure we get a boolean numpy array
+            result_np = result.to_numpy().astype(bool)
+            return BooleanArray(result_np, np.zeros(len(result),
    dtype=bool))
+
+        # Slow path: handle nulls
+        result_np = result.to_numpy().astype(bool)
+        mask = result.is_null().to_numpy()
+        return BooleanArray(result_np, mask)

    def _evaluate_op_method(self, other, op, arrow_funcs):
        pc_func = arrow_funcs[op.__name__]
```

*Figure 17.* **Left:** Expert patch on instance pandas-dev__pandas-50524 optimizing a workload via avoiding a conversion to object dtype (20.5× speedup). **Right:** CLAUDE 3.7 SONNET (OPENHANDS) instead identifies a different fast path optimization when no null elements are present, but only achieves a 2.3× speedup (scoring a speedup ratio of 0.113×).

```
--- a/pandas/core/series.py
+++ b/pandas/core/series.py
@@ -1818,7 +1818,7 @@ def to_dict(self, into: type[dict] = dict)
    -> dict:
        else:
            # Not an object dtype => all types will be the same
    so let the default
            # indexer return native python type
-            return into_c((k, v) for k, v in self.items())
+            return into_c(self.items())

    def to_frame(self, name: Hashable = lib.no_default) ->
    DataFrame:
            """
```

```
--- a/pandas/core/series.py
+++ b/pandas/core/series.py
@@ -1816,9 +1816,18 @@ class Series(base.IndexOpsMixin, NDFrame)
    :  # type: ignore[misc]
        if is_object_dtype(self):
            return into_c((k, maybe_box_native(v)) for k, v in
    self.items())
        else:
-            # Not an object dtype => all types will be the same
    so let the default
-            # indexer return native python type
-            return into_c((k, v) for k, v in self.items())
+            # Not an object dtype => use numpy fastpath to
    produce native python types
+            # by converting the underlying values to a python
    list in C and zipping
+            # with the index. This reduces Python-level boxing
    overhead.
+            values = getattr(self, "_values", None)
+            if values is None:
+                return into_c((k, v) for k, v in self.items())
+            try:
+                list_vals = values.tolist()
+            except Exception:
+                # fallback to generic iteration
+                list_vals = [v for v in values]
+            return into_c(zip(self.index, list_vals))

    def to_frame(self, name: Hashable = lib.no_default) ->
    DataFrame:
            """
```

*Figure 18.* **Left:** Expert edit on pandas-dev__pandas-50089, optimizing Series.to_dict by replacing a generator of (k, v) pairs with the view self.items(), eliminating per-element tuple allocation. **Right:** GPT-5 MINI (OPENHANDS) converts the underlying array to a Python list and zips with the index to reduce Python-level boxing during iteration (achieving a 1.98× speedup vs. the expert's 1.38×).

```
diff --git a/lib/matplotlib/transforms.py b/lib/matplotlib/
    transforms.py
index c3d1b79291..f85025efff 100644
--- a/lib/matplotlib/transforms.py
+++ b/lib/matplotlib/transforms.py
@@ -1992,6 +1992,11 @@ class Affine2D(Affine2DBase):
         self.invalidate()
         return self

+    # Cache for rotation matrices
+    _rotation_cache = {}
+    _last_theta = None
+    _last_rotate_mtx = None
+
    def rotate(self, theta):
        """
        Add a rotation (in radians) to this transform in place.
@@ -2000,10 +2005,17 @@ class Affine2D(Affine2DBase):
        calls to :meth:`rotate`, :meth:`rotate_deg`, :meth:`
    translate`
        and :meth:`scale`.
        """
-       a = math.cos(theta)
-       b = math.sin(theta)
-       rotate_mtx = np.array([[a, -b, 0.0], [b, a, 0.0], [0.0,
    0.0, 1.0]],
-                             float)
+       # Cache the rotation matrix for repeated calls with the
    same angle
+       if theta == self._last_theta and self._last_rotate_mtx is
     not None:
+           rotate_mtx = self._last_rotate_mtx
+       else:
+           a = math.cos(theta)
+           b = math.sin(theta)
+           rotate_mtx = np.array([[a, -b, 0.0], [b, a, 0.0],
    [0.0, 0.0, 1.0]],
+                                 float)
+           self._last_theta = theta
+           self._last_rotate_mtx = rotate_mtx
+
        self._mtx = np.dot(rotate_mtx, self._mtx)
        self.invalidate()
        return self
```

```
diff --git a/lib/matplotlib/transforms.py b/lib/matplotlib/
    transforms.py
index c3d1b7929128..b0456773f6a4 100644
--- a/lib/matplotlib/transforms.py
+++ b/lib/matplotlib/transforms.py
@@ -2002,9 +2002,16 @@ def rotate(self, theta):
        """
        a = math.cos(theta)
        b = math.sin(theta)
-       rotate_mtx = np.array([[a, -b, 0.0], [b, a, 0.0], [0.0,
    0.0, 1.0]],
-                             float)
-       self._mtx = np.dot(rotate_mtx, self._mtx)
+       mtx = self._mtx
+       # Operating and assigning one scalar at a time is much
    faster.
+       (xx, xy, x0), (yx, yy, y0), _ = mtx.tolist()
+       # mtx = [[a -b 0], [b a 0], [0 0 1]] * mtx
+       mtx[0, 0] = a * xx - b * yx
+       mtx[0, 1] = a * xy - b * yy
+       mtx[0, 2] = a * x0 - b * y0
+       mtx[1, 0] = b * xx + a * yx
+       mtx[1, 1] = b * xy + a * yy
+       mtx[1, 2] = b * x0 + a * y0
        self.invalidate()
        return self
```

*Figure 19.* **Left:** Expert patch on instance `matplotlib__matplotlib-22108` optimizing a rotation transform wrkload, avoiding `numpy` arithmetic overhead by operating and assigning one scalar at a time (1.9× speedup). **Right:** CLAUDE 3.7 SONNET (OPENHANDS) instead identifies a last rotation caching mechanism, and achieves a 2.4× speedup (scoring a speedup ratio of 1.292×).

