# OpenReview forum: "SWE-fficiency: Can Language Models Optimize Real-World Repositories on Real Workloads?"
_ICML.cc/2026/Conference — ICML 2026 regular_

### Official Review · Reviewer_PFFQ · 2026-03-01

**Soundness:** 4
**Presentation:** 4
**Significance:** 4
**Originality:** 3
**Overall Recommendation:** 5
**Confidence:** 4

**Summary:**

This paper introduces SWE-FFICIENCY, a repo-level benchmark for evaluating LLM-based performance engineering. The benchmark comprises 498 tasks across nine widely used data-science, machine-learning, and HPC repositories, where agents are given the full codebase and a separate performance workload, and must produce code changes that can speed up the workload while preserving correctness. The evaluation shows that there is still significant room for improvement.

**Compliance With Llm Reviewing Policy:**

Affirmed.

**Key Questions For Authors:**

1. Can you clarify how the coverage-guided test selection is done during the evaluation time? Will it be tailored to candidate LLM edits?

2. What do you envision will be changed / done if switching to Rust or other programming languages?

3. While GSO / SWE-Perf is discussed, a more explicit comparison / controlled comparison on some overlapping instances would sharpen the contribution  / better motivate the paper.

**Limitations:**

yes

**Strengths And Weaknesses:**

1. Table 1 is really nice because it clearly shows the difference between this work and prior work. Very good comparison, and I appreciate the novelty that SWE-FFICIENCY separates performance evaluation workloads from repo correctness tests.

2. This paper works on an important, unresolved, and significant problem.

3. The paper's presentation is really clear and easy to understand.

4. The experiment results are comprehensive, with metrics carefully designed and well-explained.

---

> ### Author Rebuttal · Authors · 2026-03-31
>
> We thank the reviewer for the positive evaluation of our work and highlighting our work’s positioning, presentation, comprehensiveness, and design. We address your key questions below:
>
> ---
>
> ***On coverage guided test selection details:***
>
> We would like to clarify that **coverage-guided test selection is performed during dataset construction**, rather than at evaluation time. As discussed in Appendix E.3 (Lines 986-1005), during our pipeline's third stage, we identify the existing unit tests that are specifically relevant to the codebase state and the optimization performed by the original expert. This static set of "guarding" tests is packaged with the codebase state into a benchmark task and executed for every candidate LLM edit to ensure functional correctness is preserved.
>
> This unit test selection criteria are intentionally broad to ensure high-fidelity regression testing. Rather than using a narrow line-by-line diff intersection between line coverage from tests and the expert edit git patch, we align dynamic coverage data with the **lines, functions, classes, and modules** modified in the original expert PR. We select **all repository unit test files** that then intersect these symbol definitions or function bodies. This change-focused selection helps us identify the core set of *relevant* correctness-guarding tests without requiring us to always run the *entire* test suite for a repo.
>
> ---
>
> ***On extension to Rust or other programming languages:***
>
> We agree that evaluating agents on lower-level languages like Rust or C++ is a vital next step. As noted in our limitations (Line 424), SWE-fficiency focuses on the **Scientific Python stack**. We chose this ecosystem because it represents the primary "glue" between high-level algorithmic logic and low-level performance-critical implementations (often via Cython or C extensions).
>
> Crucially, our benchmark's open-ended task formulation does not strictly prohibit other languages. An agent could technically choose to rewrite a bottleneck in Rust or C++, provided it maintains the existing Python bindings and passes the functional test harness. However, we identify two primary hurdles for a full-scale expansion of our automated construction pipeline to non-Python repositories:
>
> * **Static Analysis & Coverage:** Generalizing our coverage-guided test selection and static analysis tools to compiled languages requires significantly more complex instrumentation compared to the standardized `pytest` and `coverage-py` ecosystem in Python.
> * **Environment Heterogeneity:** Unlike the relatively unified `pip`/`conda` build process, compiled language repositories often feature highly varied build toolchains and dependency management systems, making automated environment "sandboxing" more difficult to scale.
>
> By targeting the foundational scientific libraries first, we provide a rigorous testbed for optimization reasoning in a domain where the impact of performance gains is immediate and well-documented.
>
> ---
>
> **On the relation to GSO and SWE-Perf:**
>
> While GSO and SWE-Perf also explore repository-level optimization, SWE-fficiency addresses a distinct and more challenging dimension of performance engineering: exploratory bottleneck discovery (Section 1, Lines 35-70; Table 1, Lines 110-125). While prior benchmarks focus on *local* optimization performance, SWE-fficiency evaluates the agent's ability to reason about the *global* interaction between code, workloads, and existing test infrastructure.
>
> * **The "Discovery Gap"**: Unlike SWE-Perf, which provides agents with specific target functions to optimize, or GSO, which provides LM-generated oracle correctness scripts, SWE-fficiency requires agents to exercise further skills in repository navigation and discovery during performance optimization. Agents must use expert-annotated workloads to both independently identify bottlenecks and also discover relevant in-repo unit tests to help self-verify during optimization.
> * **Strict Semantic Invariants (Pass-to-Pass)**: Our unique "zero-test-change" PR selection ensures a "pass-to-pass" evaluation. This provides a high-fidelity signal that the optimization is truly a refactor of the implementation (improving runtime without altering any functional edge cases) rather than a side-effect of changing the test's scope.
> * **Concurrent Development:** SWE-fficiency, GSO, and SWE-Perf were developed concurrently during 2025. As such, our work represents an independent and complementary approach.
>
> We believe this distinction, moving from "guided optimization" to "autonomous performance engineering," is an important contribution to the community.

---

> > ### Author Rebuttal · Reviewer_PFFQ · 2026-04-02
> >
> > Thanks for the clarification.

---

### Official Review · Reviewer_ED4Y · 2026-03-05

**Soundness:** 3
**Presentation:** 3
**Significance:** 2
**Originality:** 3
**Overall Recommendation:** 4
**Confidence:** 4

**Summary:**

This paper introduces **SWE-fficiency**, a repository-level benchmark for **workload-specific performance optimization** in real-world Python scientific/HPC libraries. Each task provides a full codebase and a slow workload; an LM agent must (i) investigate runtime behavior, (ii) localize bottlenecks and relevant unit tests, and (iii) submit a patch that preserves correctness while achieving speedups comparable to (or exceeding) an expert human patch mined from merged GitHub PRs. The authors contribute an automated+human-in-the-loop pipeline to scrape and validate performance PRs (including test localization in the common “no new tests added” regime), define a normalized **Speedup Ratio** metric against expert baselines, and evaluate modern agent systems, finding they recover only a small fraction of expert speedups and often fail via mis-localization, shallow edits, or correctness breakage.

**Compliance With Llm Reviewing Policy:**

Affirmed.

**Final Justification:**

The authors have well addressed my concerns.

**Key Questions For Authors:**

1. For a subset of tasks, can the authors provide *multiple* workloads (or perturbed workload variants) and report how often high-scoring patches retain speedups under distributional shifts (e.g., larger N, different sparsity, different dtypes)?
2. Do you have evidence on how often an agent patch that passes the localized/unit tests and improves the provided workload nevertheless causes slowdowns on other related code paths? Even a small “audit” study would help.
3. Have you considered adding a lightweight memory constraint or auxiliary reporting (peak RSS / allocations) to flag extreme memory-for-runtime solutions? If not, could you quantify how often “cache-heavy” solutions appear among top agent submissions?
4. How sensitive are rankings to measurement choices such as using median vs mean, number of repeats, warmup strategy, or alternative harnesses (e.g., pyperf/asv-style calibration)? A brief ablation would strengthen confidence for close model comparisons.
5. Can you characterize what fraction of tasks correspond to performance-critical APIs vs higher-level  functions, and whether agent performance differs systematically across these categories?

**Limitations:**

Yes

**Strengths And Weaknesses:**

### Strengths

- **Important evaluation target.** The benchmark focuses on *how-to-fix for efficiency* (not just functional correctness), which is increasingly relevant as LLMs are deployed in software maintenance.
- **Task formulation captures genuine SWE complexity.** Requiring agents to navigate a full repository, identify relevant tests, and preserve correctness while optimizing runtime is meaningfully harder than function-level code optimization tasks.
- **Practical evaluation engineering.** The paper makes a clear effort toward reproducibility (fixed environment, isolation measures, repeated measurements) and includes failure-mode analyses that are actionable for future work.


### Weaknesses

- **Single-workload, runtime-only objective risks overfitting.** A patch can be *correct* and score well by exploiting idiosyncrasies of the provided workload scale while providing limited general benefit (or causing regressions on other inputs/paths). This is not merely “reward hacking”; it is a fundamental ambiguity in what constitutes a “useful” optimization in practice.
- **Limited coverage of real performance trade-offs.** In real maintenance workflows, engineers routinely consider *multiple workloads* and constraints (e.g., memory footprint, latency vs throughput, worst-case behavior, or regressions on other APIs). A single workload and runtime-only scoring may under-represent this decision-making and could implicitly reward undesirable space–time trade-offs in some cases.
- **External validity remains somewhat unclear.** While the benchmark is excellent for measuring *workload-specific bottleneck fixing*, it is less clear how leaderboard gains translate to broader performance engineering competence (e.g., generalizable speedups, avoiding regressions across a realistic workload suite).

---

> ### Author Rebuttal · Authors · 2026-03-31
>
> We thank the reviewer for the detailed review and for recognizing our efforts to make our performance benchmark reproducible, reliable, and reflective of practical performance engineering. Below, we address your concerns
>
> ---
>
> **On single-workload, runtime-only design choices:**
>
> We agree that comprehensive performance engineering involves global workloads and complex regression trade-offs. However, SWE-fficiency is intentionally designed to evaluate a specific, prerequisite skill: **repository-level, targeted runtime bottleneck optimization** (Lines 86–94).
>
> This is a fundamental building block; an agent must first demonstrate the ability to speed up a specific workload before it can manage the multi-objective trade-offs of a production environment. Just as SWE-bench isolates bug fixing without requiring agents to manage complete software engineering lifecycle (i.e. software deployment or long-term feature roadmap planning), SWE-fficiency isolates the skill of workload-specific runtime optimization within broader performance engineering. Targeting this isolated capability allows us to reason about specific failure modes, such as function-level mislocalization (Figure 6, Line 340). While future work could include broader performance suites, the current "unsaturated" performance on single workloads ($\< 0.23\\times$ expert speedup) suggests agents still have much to improve on this foundational runtime reasoning task.
>
> ---
>
> ***On adding memory constraints and “cache-heavy” solutions:***
>
> We did not make memory constraints a focus for two primary reasons:
>
> 1. **Alignment with Expert Intent:** Our data collection pipeline demonstrates that, in the repos we focus on, developers focus on runtime optimizations over memory optimization by a significant margin. Using keyword matching on PR metadata as done in Stage 2 (Lines 107-135), runtime-related PRs (11.9% of total PR count) are **7.1x** more common than memory-related PRs (1.7%). In our annotation process, we found that maintainers overwhelmingly accepted performance PRs based on the runtime speedups documented in the descriptions. Conversely, formal memory regression tests were absent in nearly all of these instances. Evaluating agents against memory overhead limits would therefore impose a standard that human experts did not apply to the original solutions.
> 2. **Subjectivity of "Cache-Heavy" Solutions:** Caching is often repository-dependent and hard to enforce in a standard way. For example, sympy employs a global results cache and views broad memoization as an acceptable trade-off ([link](https://github.com/sympy/sympy/blob/master/sympy/core/cache.py#L38-L42)). During annotation of sympy workloads, we cleared caches between timing runs to benchmark the desired behavior. This view is not shared in pandas or numpy, where caching computation is less preferred ([pandas link](https://github.com/pandas-dev/pandas/issues/50547), [numpy link](https://stackoverflow.com/questions/79391458/surprising-lack-of-speedup-in-caching-numpy-calculations)).
>
> ---
>
> ***On sensitivity of benchmark measurement and model comparisons:***
>
> Measurement robustness is central to our benchmark design. Our benchmark metric is highly stable, exhibiting a variance of $\< 0.5\\%$ of the raw score across independent runs **(Appendix N, Line 1870).** This was achieved through:
>
> * **Hardware Isolation (Appendix F):** We pin each worker to exclusive physical CPU cores and memory resources to prevent resource contention.
> * **Workload Tuning:** During annotation, we ensure each workload uses a number of repeats (up to $10^6$) such that measurement noise is minimized and measured expert speedups are statistically significant.
>
> ---
>
> ***On whether tasks correspond to performance-critical APIs:***
>
> We recognize that "performance-criticality" is subjective and varies by use case. Our selection process was designed not to capture every possible performance scenario, but to focus on **representative optimization patterns** within a set of foundational libraries of the Python ecosystem (e.g., NumPy, SciPy, pandas). These repositories were specifically chosen because they are core scientific infrastructure for thousands of downstream applications, making optimizations within them high-leverage by definition.
>
> Rather than using synthetic bottlenecks, our tasks are derived from merged runtime optimization PRs. This ensures the workloads represent code paths that the original library maintainers (the domain experts) deemed significant enough to optimize and merge into the main codebase.
>
> While we do not claim our results generalize to all software domains, they provide a rigorous look at how agents handle real-world, expert-vetted performance issues in high-impact libraries. We appreciate the point regarding task representativeness and will update the limitations section in our paper to further emphasize this framing.

---

> > ### Author Rebuttal · Reviewer_ED4Y · 2026-04-03
> >
> > Thanks for your detailed response. My concerns have been well addressed.

---

### Official Review · Reviewer_YW5p · 2026-03-06

**Soundness:** 4
**Presentation:** 4
**Significance:** 4
**Originality:** 3
**Overall Recommendation:** 5
**Confidence:** 4

**Summary:**

The authors propose SWE-fficiency, benchmarking LLMs on repository-level workload performance optimization problems. SWE-fficiency implements a data pipeline to identify performance-related issues and further curate performance-oriented evaluator script for each individual issues *manually*. The authors show interesting and meaningful results in the experiment section.

**Compliance With Llm Reviewing Policy:**

Affirmed.

**Final Justification:**

The authors clarified my questions. I keep my current score.

**Key Questions For Authors:**

1. Why $\$1$ threshold for the agentic scaffold? I do understand the resource limitations, but I'd like to see how budget influences the speedup in a subset of the dataset with a cheaper model.
2. In figure 8 there is a comparison between human's code and agent's code. Can you provide the reasoning of the LLM for the agent's code?

**Limitations:**

Yes.

**Strengths And Weaknesses:**

Thanks for submitting to ICML 2026. The paper reads well and demonstrates great amount of work to build this new dataset.

### Soundness

The authors construct semi-automatic pipeline to curate the dataset, and measure the speedup with measurable significance. The approaches in the paper are sound.

### Presentation

The paper is in good shape. Easy to follow. The authors need to fix the conclusion section title (instead of using paragraph command) to stay uniform with the academic paper standards.

### Significance

The benchmark dataset introduces performance oriented tests, which provides the *right* oracle for the task, instead of using correctness checking test cases. The manual efforts help researchers get *the* sensible benchmark results for LLMs and agents. The problem of harnessing agents on performance optimization tasks is important as the correctness fixing problem.

### Originality

Compared to prior work (SWE-perf), SWE-fficiency introduces performance tests for individual task instead of using the existing unit test cases designed for correctness checking. The authors discuss several observations in agent trajectories on SWE-fficiency tasks and reveal limitations of the existing agents on performance optimization tasks.

---

> ### Author Rebuttal · Authors · 2026-03-31
>
> We sincerely thank the reviewer for their thoughtful feedback and the positive assessment of our work, highlighting the dataset construction soundness, clarity of presentation, and overall significance of our work. Below we address your key questions:
>
> ---
>
> > “Why $1 threshold for the agentic scaffold? I do understand the resource limitations, but I'd like to see how budget influences the speedup in a subset of the dataset with a cheaper model.”*
>
> We clarify that the **$1 token-spend limit was only for the SWE-agent scaffold**. All of the primary OpenHands harness results shown in Table 2 and 3 were conducted **without any dollar cost budget limit**, constrained only by a 100-action turn cap and a 3-hour wall-clock limit per task, identical to the inference settings conducted in GSO [1].
>
> ---
>
> > “In figure 8 there is a comparison between human's code and agent's code. Can you provide the reasoning of the LLM for the agent's code?”*
>
> We do not have access to the explicit internal "thinking traces" or monologues for these specific runs, as neither the models used nor the OpenHands harness were configured to log them at the time. However, we analyzed the agent's reasoning post-hoc by examining its **action trajectories**: the specific sequence of profiling, testing, and incremental edits it performed. This informed the content of Section 4.2 and the captions for Figures 8 and 9\.
>
> From these trajectories, we observe a clear **"convenience bias"** (Lines 271-315). In Figure 8, the agent’s reasoning, as evidenced by its actions, was to identify the immediate bottleneck and implement a localized "fast-path" (exiting early for non-null elements). While this achieved a 2.3x speedup, it represents a "satisficing" behavior where the agent stops once a measurable improvement is secured.
>
> In contrast, the human expert’s 20.5x speedup was achieved through systemic architectural restructuring—specifically, removing object datatype conversion overhead. Agents commonly employed bash profiling tools like `cProfile`, and in many cases, identified the location of a superficial bottleneck, but failed to reason toward the deeper, systemic cause. To support further community analysis of these decision-making processes, we plan to open-source all agent trajectories alongside our dataset and benchmark harness.
>
> ---
>
> [1] Shetty, Manish, et al. "GSO: Challenging Software Optimization Tasks for Evaluating SWE-Agents." *Advances in Neural Information Processing Systems*, vol. 38, 2025\. [https://doi.org/10.48550/arXiv.2505.23671](https://doi.org/10.48550/arXiv.2505.23671).

---

> > ### Author Rebuttal · Reviewer_YW5p · 2026-04-02
> >
> > Thanks for the clarifications. I would like to keep the score.

---

### Official Review · Reviewer_7miS · 2026-03-12

**Soundness:** 3
**Presentation:** 2
**Significance:** 4
**Originality:** 3
**Overall Recommendation:** 4
**Confidence:** 3

**Summary:**

The paper constructs a benchmark SWE-FFICIENCY to evaluate the ability of LM agents to perform real-world repo-level optimizations. The benchmark consists of 498 tasks curated from nine Python repositories. It requires agents to understand code semantics, localize performance bottlenecks, and generate patches that improve performance while maintaining correctness. Evaluations yield various interesting findings, such as frontier models significantly underperform human experts, LM agents often fail to identify the location of performance bottlenecks and LM agents tend to take shortcut and make satisficing optimizations.

**Compliance With Llm Reviewing Policy:**

Affirmed.

**Final Justification:**

The rebuttal addressed my main concerns. Some minor concerns still remain, e.g., the authors did not explain what static analysis tools are used and how they are used. Plus, in the LM-based workload generation evaluation the setting is unfair, since 1) Gemini-2.5-flash is not a frontier model so it remains unknown whether stronger models could produce better workloads than human, 2) in LM-based workload generation, some key information such as PR descriptions are not included, while they are visible to human annotators.

**Key Questions For Authors:**

1. The paper found that LM agents often miss the correct location to optimize. Could you perform a correlation analysis on whether the correct bottleneck is found and whether the code has been successuflly optimized to confirm that?
2. In Figure 3, a lot of tasks have runtime of 10-3 s or smaller scale. How stable is the evaluation of these tasks?

**Limitations:**

Yes

**Strengths And Weaknesses:**

Strengths:
- Timely topic. LM agent code performance optimization is an important topic and good benchmarks are highly necessary.
- Solid benchmarking methodology and framework. The authors design a rigorous data curation pipeline to obtain a high-quality set of performance optimization tasks. The authors employed techniques including Docker containeralization and CPU pinning to improve performance reproducibility.
- Solid evaluatin and insightful findings. The authors deeply investigate the performance of LM agents and human experts and found issues of LM agents like mislocalization and taking shortcut.

Weaknesses:
- Some findings have been found in existing literature, e.g., [1] found LM agents tend to be lazy and fail to precisely localize performance bottleneck.
- Manual annotation limits the scalability of the approach. While the data collection pipeline is mostly automated, Stage IV relies heavily on manual annotation by the authors to create the performance workloads, which could be expensive and error-prune.
- The writing of some sentences is not easy to follow, e.g., "Another LM pattern is to bake benchmark properties into patches, producing impressive but brittle wins."
- Correctness harness too weak. The pipeline ensures at least one correctness test cover the performance-improving edit, but is one correctness test enough to harness correctness? With weak harnesses, LLMs could easily pass generate false-positive patches.
- Only speedup statitistics are reported, whereas the improvement in percentage is not included. Although the evaluation reliles on the harmonic average which can mitigate the dominating effect of large values, it is sensitive to small values. The paper did not discuss that potential threat.

[1] Shetty, M., Jain, N., Liu, J., Kethanaboyina, V., Sen, K., and Stoica, I. Gso: Challenging software optimization tasks for evaluating swe-agents, 2025. URL https://arxiv.org/abs/2505.23671.

---

> ### Author Rebuttal · Authors · 2026-03-31
>
> We thank the reviewer for the high-confidence accept recommendation and for recognizing our contributions. We will improve the writing clarity in the updated paper.
>
> **On relation to GSO:**
>
> While our work aligns with GSO regarding localization difficulty, we provide a complementary and deeper diagnosis of the agent-expert gap.
>
> * By treating expert speedup as "mass" distributed across functions, we use Expert-Relative Coverage (ERC) to show that agents miss 68%–75% of achievable speedup primarily by editing non-bottleneck functions, even when in the correct file (Section 4.1, Figure 6, Appendix I).
> * Crucially, expert patch size does not predict task-level performance (Fig 4), suggesting that localization precision, not just code volume, is a barrier.
> * Unlike GSO providing oracle scripts for verifying correctness, we require agents to localize existing repository unit tests, mimicking a real-world "library maintainer" workflow .
>
> ---
>
> **On manual annotation and scalability:**
>
> Manual annotation is inherent in building many high quality benchmarks \[1, 2, 3\].
>
> * **Expert intention:** PR authors write bespoke performance scripts in their PR descriptions that represent the specific "targeted optimization intent" for their changes (Appendix E.4, Lines 1007–1090). We annotate this intent into standardized workloads, allowing us to fairly score agents against expert speedup. Every annotation was agreed on by two closely-calibrated author annotators.
> * **Reliability:** Pilot study showed LM-synthesized workloads (Appendix K) were less reliable than our annotations; 47% of synthetic workloads failed to show gains after applying expert patches. Manual annotations showed more differentiating improvements 76% of the time.
> * **Scalability (Appendix E):** Our automated filters prune over 99% of candidates (100k PRs down to \~1k) before manual intervention. We view "LMs-in-the-loop" as promising future work to further increase yield.
>
> ---
>
> **On correctness harness strength:**
>
> We agree that a strong correctness harness is particularly important. The "at least one test" rule is a minimum inclusion threshold, not the total harness. In practice, the median task is guarded by 7,723 tests, over 90% of tasks (449/498) are protected by more than 100 tests, and ~74% have more than 1,000 tests. There is only one instance that has only one test (`sympy__sympy-10919`) but consists of multiple assertions across a wide range of inputs ([link](https://github.com/jtnydv25/sympy/blob/dcb6b43/sympy/ntheory/tests/test_partitions.py)).
>
> ---
>
> **On metric sensitivity:**
>
> While the harmonic mean is standard for speedup ratios, we have updated our aggregation to impose a *lower bound of 0.001* on all per-instance Speedup Ratio (SR) values. This penalizes low-scoring patches without allowing near-zero values to obscure overall capabilities. Opus 4.5 leads at 0.225x, followed by GPT-5 (0.157x) and 4.6 Opus (0.155x).
>
> ---
>
> **On correlation: Localization vs. Optimization Success**
>
> We find that the inability to pinpoint precise functional bottlenecks, rather than the volume of code changes prevents them from achieving expert parity. We performed the reviewer’s requested correlation by computing Expert-Relative Coverage (ERC) (Figure 6 and Appendix I.3), which quantifies the overlap between agent edits and the specific functions responsible for expert gains.
>
> By treating expert speedup as "mass" distributed across expert-edited functions, ERC metrics measure how much expert speedup was in the file or function that an LM chose to edit. We found agents miss *68.6%–75.4%* of achievable expert speedup primarily due to functional mislocalization. While agents frequently identify the correct file (>55% overlap), they consistently target non-bottleneck functions. This suggests that *localization precision*, not the volume of code changed, is the primary barrier to matching expert performance.
>
> ---
>
> **On stability of performance measurement:**
>
> We agree that the stability of performance measurement is extremely important. As shown in Table 11 (Appendix N, Line 1870), the variance in our metric across independent runs is negligible (*< 0.5% of raw score*), confirming that measurement noise does not affect the relative ranking of agent capabilities. We achieved this via:
>
> * **Workload tuning during annotation:** During annotation (Sec 2.1), we tailored the iterations for each workload to ensure a stable runtime distribution. For sub-millisecond tasks, we increased timeit.repeat up to 10^6 iterations to effectively eliminate the impact of micro-variance.
> * **Hardware isolation:** We pin evaluation workers to specific CPU cores and memory nodes (Appendix F.2, Line 1172). These measures prevent core-level interference and background system jitter from interfering with precise timings.
>
> [1] Terminal-Bench https://arxiv.org/abs/2601.11868
>
> [2] AlgoTune https://arxiv.org/abs/2507.15887
>
> [3] https://openai.com/index/introducing-swe-bench-verified/

---

> > ### Author Rebuttal · Reviewer_7miS · 2026-04-02
> >
> > Thanks the authors for addressing the issues. I have the below remaining concern.
> >
> > **On correctness harness strength:**
> > I was asking about number of tests covering the patched location, not the total number of executed tests. There could be thousands of executed tests but only one test covers the patched location. Weak harness means some aggressive, over-optimization patch, or even fundamentally wrong patch could pass the correctness check and produce false high scores. Especially in swefficiency setting the workload is visible to the agent. So I don't see any harness or efforts that could prevent agents from cheating.

---

> > > ### Author Response · Authors · 2026-04-02
> > >
> > > We are grateful that our rebuttal has addressed your concerns. We clarify regarding your remaining concern below:
> > >
> > > ***On correctness harness strength:***
> > >
> > > The statistics we provided previously in our first rebuttal actually **refer to intersecting tests**, specifically unit-tests that execute at least one line of the expert optimization patch or intersect the function/class body that the expert patch modifies. As detailed in Appendix C.3 (Line 986), we use static analysis alongside coverage to filter the full test suite to identify tests that touch (and protect) the specific code changes that the expert made.
> > >
> > > Thus, to clarify, the median of 7,723 tests is not the total repo count, but the number of tests specifically covering the expert patched location. As mentioned before, over 90% of tasks are guarded by more than 100 intersecting tests, ensuring robust coverage while also ensuring the entire benchmark can run in reasonable time. We also emphasize that all base PRs that were used for SWE-fficiency benchmark tasks come from approved and merged performance optimization PRs in the repos we scrape.

---

### Decision · Program_Chairs · 2026-04-30

**Decision:**

Accept (regular)

**Comment:**

The paper introduces a benchmark of almost 500 real-world tasks drawn from 9 widely used Python scientific/HPC libraries that requires agents to localize performance bottlenecks, find relevant tests and produce patches that match or exceed expert speedups while preserving correctness. The problem is well motivated, and the reviewers all agreed that packaging real merged performance PRs into a benchmark fills a clear gap in the field. The exposition is generally clear and the technical contributions (the datapipeline, isolation for reproducible profiling, ERC localization analysis) are solid. Reviewers raised concerns about the scoring (using single workloads), manual annotation steps and the need for more detail on static analysis/test selection tooling. During the discussion phase, the authors directly addressed these concerns in their rebuttal by clarifying that reported counts refer to tests that intersect the patched locations, explained the pipeline's automated filtering and why some manual annotation was necessary. The reviewers acknowledged that their concerns were resolved, and praised the benchmark as a substantial, reproducible contribution to further research on automated performance engineering.